# A unifying framework for mean-field theories of asymmetric kinetic Ising systems

Miguel Aguilera [1,2,3 ✉], S. Amin Moosavi[4,5] & Hideaki Shimazaki [4,6]

Kinetic Ising models are powerful tools for studying the non-equilibrium dynamics of complex systems. As their behavior is not tractable for large networks, many mean-field methods have been proposed for their analysis, each based on unique assumptions about the system's temporal evolution. This disparity of approaches makes it challenging to systematically advance mean-field methods beyond previous contributions. Here, we propose a unifying framework for mean-field theories of asymmetric kinetic Ising systems from an information geometry perspective. The framework is built on Plefka expansions of a system around a simplified model obtained by an orthogonal projection to a sub-manifold of tractable probability distributions. This view not only unifies previous methods but also allows us to develop novel methods that, in contrast with traditional approaches, preserve the system's correlations. We show that these new methods can outperform previous ones in predicting and assessing network properties near maximally fluctuating regimes.

[1] IAS-Research Center for Life, Mind, and Society, Department of Logic and Philosophy of Science, University of the Basque Country, Donostia, Spain. [2] Department of Informatics & Sussex Neuroscience, University of Sussex, Falmer, Brighton, UK. [3] ISAAC Lab, Aragón Institute of Engineering Research (I3A), University of Zaragoza, Zaragoza, Spain. [4] Graduate School of Informatics, Kyoto University, Kyoto, Japan. [5] Present address: Department of Neuroscience, Brown University, Providence, RI, USA. [6] Present address: Center for Human Nature, Artificial Intelligence, and Neuroscience (CHAIN), Hokkaido University, Sapporo, Hokkaido, Japan. ✉email: sci@maguilera.net

Advances in high-throughput data acquisition technologies for very large biological and social systems are providing unprecedented possibilities to investigate their complex, non-equilibrium dynamics. For example, optical recordings from genetically modified neural populations make it possible to simultaneously monitor activities of the whole neural network of behaving C. elegans[1] and zebrafish[2], as well as thousands of neurons in the mouse visual cortex[3]. Such networks generally exhibit out-of-equilibrium dynamics[4], and are often found to self-organize near critical regimes at which their fluctuations are maximized[5,6]. Evolution of such systems cannot be faithfully captured by methods assuming an asymptotic equilibrium state. Therefore, in general, there is a pressing demand for mathematical tools to study the dynamics of large-scale, non-equilibrium complex systems and to analyze high-dimensional datasets recorded from them.

The kinetic Ising model with asymmetric couplings is a prototypical model for studying such non-equilibrium dynamics in biological[7,8] and social systems[9]. It is described as a discrete-time Markov chain of interacting binary units, resembling the nonlinear dynamics of recurrently connected neurons. The model exhibits non-equilibrium behavior when couplings are asymmetric or when model parameters are subject to rapid changes, ruling out quasi-static processes. These conditions induce a time reversal asymmetry in dynamical trajectories, leading to positive entropy production (the second law of thermodynamics) as revealed by the fluctuation theorem[10–15] (see refs. [16,17] for reviews). This time-asymmetry is characteristic of non-equilibrium systems as it can only be displayed by systems in which energy dissipation takes place[18]. In the case of symmetric connections and static parameters, the model converges to an equilibrium stationary state. Consequently, it is a generalization of its equilibrium counterpart known as the (equilibrium) Ising model[19].

The forward Ising problem refers to calculating statistical properties of the model, such as mean activation rates (mean magnetizations of spins) and correlations, given the parameters of the model. In contrast, inference of the model parameters from data is called the inverse Ising problem[20]. In this regard, kinetic Ising models[21,22] and their equilibrium counterparts[23–25] have become popular tools for modeling and analyzing biological and social systems. In addition, they capture memory retrieval dynamics in classical associative networks. Namely, they are equivalent to the Boltzmann machine, extensively used in machine learning applications[20]. Unfortunately, exact solutions of the forward and inverse problems often become computationally too expensive due to the combinatorial explosion of possible patterns in large, recurrent networks or the high volume of data, and applications of exact or sampling-based methods are limited in practice to around a hundred of neurons[5,25,26]. In consequence, analytical approximation methods are necessary for analysing large systems. In this endeavour, mean-field methods have emerged as powerful tools to track down otherwise intractable statistical quantities.

The standard mean-field approximations to study equilibrium Ising models are the classical naive mean-field (nMF) and the more accurate Thouless-Anderson-Palmer (TAP) approximations[27]. These methods have also been employed to solve the inverse Ising problem[28–31]. Plefka demonstrated that the nMF and TAP approximations for the equilibrium model can be derived using the power series expansion of the free energy around a model of independent spins, a method which is now referred to as the Plefka expansion[32]. This expansion up to the first and second orders leads to the nMF and TAP mean-field approximations respectively. The Plefka expansion was later formalized by Tanaka and others in the framework of information geometry[33–37].

In non-equilibrium networks, however, the free energy is not directly defined, and thus it is not obvious how to apply the Plefka expansion. Kappen and Spanjers[38] proposed an information geometric approach to mean-field solutions of the asymmetric Ising model with asynchronous dynamics. They showed that their second-order approximation for an asymmetric model in the stationary state is equivalent to the TAP approximation for equilibrium models. Later, Roudi and Hertz derived TAP equations for nonstationary states using a Legendre transformation of the generating functional of the set of trajectories of the model[39]. Another study by Roudi and Hertz extended mean-field equations to provide expressions for the nonstationary delayed correlations assuming the presence of equal-time correlations at the previous step[40]. Yet another interesting method proposed by Mézard and Sakellariou approximates the local fields by a Gaussian distribution according to the central limit theorem, yielding more accurate results for fully asymmetric networks[41]. This method was later extended to include correlations at the previous time step, improving the results for symmetric couplings[42]. More recently, Bachschmid-Romano et al. extended the path-integral methods in ref. [39] with Gaussian effective fields[43], not only recovering ref. [41] for fully asymmetric networks but also proposing a method that better approximates mean rate dynamics by conserving autocorrelations of units. Although many choices of mean-field methods are available, the diversity of methods and assumptions makes it challenging to advance systematically over previous contributions.

Here, we propose a unified approach for mean-field approximations of the Ising model. While our method is applicable to symmetric and equilibrium models, we focus for generality on asymmetric kinetic Ising models. Our approach is defined as a family of Plefka expansions in an information geometric space. This approach allows us to unify and relate existing mean-field methods and to provide expressions for other statistics of the systems such as pairwise correlations. Furthermore, our approach can be extended beyond classical mean-field assumptions to propose novel approximations. Here, we introduce an approximation based on a pairwise model that better captures network correlations, and we show that it outperforms existing approximations of kinetic Ising models near a point of maximum fluctuations. We also provide a data-driven method to reconstruct and test if a system is near a phase transition by combining the forward and inverse Ising problems, and demonstrate that the proposed pairwise model more accurately estimates the system's fluctuations and its sensitivity to parameter changes. These results confirm that our unified framework is a useful tool to develop methods to analyze large-scale, non-equilibrium biological and social dynamics operating near critical regimes. In addition, since the methods are directly applicable to Boltzmann machine learning, the geometrical framework introduced here is relevant in machine learning applications.

The paper is organized as follows. First, we introduce the kinetic Ising model and its statistical properties of interest. Second, we introduce our framework for the Plefka approximation methods from a geometric perspective. To explain how it works, we derive the classical naive and TAP mean-field approximations under the proposed framework. Third, we show that our approach can unify other known mean-field approximation methods. We then propose a novel pairwise approximation under this framework. Finally, we compare different mean-field approximations in solving the forward and inverse Ising problems, as well as in performing the data-driven assessment of the system's sensitivity. The last section is devoted to discussion.

## Results

**The kinetic Ising model.** The kinetic Ising model is the least structured statistical model containing delayed pairwise interactions between its binary components (i.e., a maximum caliber model[44]). The system consists of $N$ interacting binary variables (down or up of Ising spins or inactive or active of neural units) $s_{i,t} \in \{-1, +1\}, i = 1, 2, \ldots, N$, evolving in discrete-time steps $t$ with parallel dynamics. Given the configuration of spins at $t-1$, $\mathbf{s}_{t-1} = \{s_{1,t-1}, s_{2,t-1}, \ldots, s_{N,t-1}\}$, spins $\mathbf{s}_t$ at time $t$ are conditionally independent random variables, updated as a discrete-time Markov chain, following

$$P(\mathbf{s}_t | \mathbf{s}_{t-1}) = \prod_i \frac{e^{s_{i,t} h_{i,t}}}{2 \cosh h_{i,t}}, \tag{1}$$

$$h_{i,t} = H_i + \sum_j J_{ij} s_{j,t-1}. \tag{2}$$

The parameters $\mathbf{H} = \{H_i\}$ and $\mathbf{J} = \{J_{ij}\}$ represent local external fields at each spin and couplings between pairs of spins respectively. When the couplings are asymmetric (i.e., $J_{ij} \neq J_{ji}$), the system is away from equilibrium because the process is irreversible with respect to time. Given the probability mass function of the previous state $P(\mathbf{s}_{t-1})$, the distribution of the current state is:

$$P(\mathbf{s}_t) = \sum_{\mathbf{s}_{t-1}} P(\mathbf{s}_t | \mathbf{s}_{t-1}) P(\mathbf{s}_{t-1}). \tag{3}$$

This marginal distribution $P(\mathbf{s}_t)$ is not factorized (except at $\mathbf{J} = \mathbf{0}$), but it rather exhibits a complex statistical structure, generally containing higher-order spin interactions. We can apply this equation recursively, e.g., decomposing $P(\mathbf{s}_{t-1})$ in terms of the distribution $P(\mathbf{s}_{t-2})$, and trace the evolution of the system from the initial distribution $P(\mathbf{s}_0)$.

In this article, we use variants of the Plefka expansion to calculate some statistical properties of the system. Namely, we investigate the average activation rates $\mathbf{m}_t$, correlations between pairs of units (covariance function) $\mathbf{C}_t$, and delayed correlations $\mathbf{D}_t$ given by

$$m_{i,t} = \sum_{\mathbf{s}_t} s_{i,t} P(\mathbf{s}_t), \tag{4}$$

$$C_{ik,t} = \sum_{\mathbf{s}_t} s_{i,t} s_{k,t} P(\mathbf{s}_t) - m_{i,t} m_{k,t}, \tag{5}$$

$$D_{il,t} = \sum_{\mathbf{s}_t, \mathbf{s}_{t-1}} s_{i,t} s_{l,t-1} P(\mathbf{s}_t, \mathbf{s}_{t-1}) - m_{i,t} m_{l,t-1}. \tag{6}$$

Note that $\mathbf{m}_t$ and $\mathbf{D}_t$ are sufficient statistics of the kinetic Ising model. Therefore, we will use them in solving the inverse Ising problem (see Methods). We additionally consider the equal-time correlations $\mathbf{C}_t$ as they are commonly used to describe neural systems, and are investigated by some of the mean-field approximations in the literature[40]. Calculation of these expectation values is analytically intractable and computationally very expensive for large networks, due to the combinatorial explosion of the number of possible states. To reduce this computational cost, we approximate the marginal probability distributions (Eq. (3)) by the Plefka expansion method that utilizes an alternative, tractable distribution.

**Geometrical approach to mean-field approximation.** Information geometry[37,45,46] provides clear geometrical understanding of information-theoretic measures and probabilistic models[15,47,48]. Using the language of information geometry, we introduce our method for mean-field approximations of kinetic Ising systems.

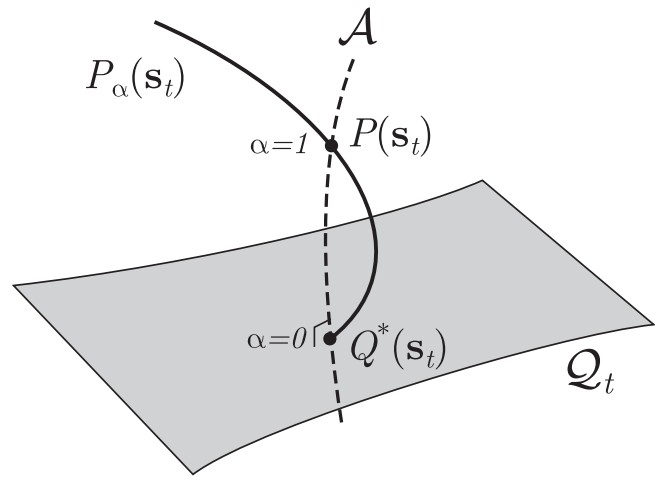

**Fig. 1 A geometric view of the approximations based on Plefka expansions.** The point $P(\mathbf{s}_t)$ is the marginal distribution of a kinetic Ising model at time $t$. The submanifold $\mathcal{Q}_t$ is a set of tractable distributions, for example a manifold of independent models. The points in $\mathcal{A}$ correspond to a m-geodesic, that is a linear mixture of $P(\mathbf{s}_t)$ and $Q^*(\mathbf{s}_t)$ on $\mathcal{Q}_t$, where for independent $\mathcal{Q}_t$ all points on $\mathcal{A}$ share the same mean values $\mathbf{m}_t$. Geometrically, $\mathcal{A}$ constitutes the m-projection from $P(\mathbf{s}_t)$ to $\mathcal{Q}_t$, defining $Q^*(\mathbf{s}_t)$ as the closest point in the submanifold $\mathcal{Q}_t$ to the point $P(\mathbf{s}_t)$[47]. The Plefka expansion is defined by expanding an $\alpha$-dependent distribution $P_\alpha(\mathbf{s}_t)$ that satisfies $P_{\alpha=0}(\mathbf{s}_t) = Q^*(\mathbf{s}_t)$ and $P_{\alpha=1}(\mathbf{s}_t) = P(\mathbf{s}_t)$.

Let $\mathcal{P}_t$ be the manifold of probability distributions at time $t$ obtained from Eq. (3). Each point on the manifold corresponds with a set of parameter values. The manifold $\mathcal{P}_t$ contains submanifolds $\mathcal{Q}_t$ of probability distributions with analytically tractable statistical properties (See Fig. 1). We use this tractable manifold, i.e., a reference model, to approximate a target point $P(\mathbf{s}_t | \mathbf{H}, \mathbf{J})$ in the manifold $\mathcal{P}_t$ and its statistical properties $\mathbf{m}_t, \mathbf{C}_t, \mathbf{D}_t$.

The simplest submanifold $\mathcal{Q}_t$ is the manifold of independent models, used in classical mean-field approximations to compute average activation rates. Each point on this submanifold corresponds to a distribution

$$Q(\mathbf{s}_t | \boldsymbol{\Theta}_t) = \prod_i \frac{e^{s_{i,t} \Theta_{i,t}}}{2 \cosh \Theta_{i,t}}, \tag{7}$$

where $\boldsymbol{\Theta}_t = \{\Theta_{i,t}\}$ is the vector of parameters that represents a point in $\mathcal{Q}_t$. This distribution does not include couplings between units, and its average activation rate is immediately given as $m_{i,t} = \tanh \Theta_{i,t}$.

Our first goal is to find the average activation rates of the target distribution $P(\mathbf{s}_t | \mathbf{H}, \mathbf{J})$. It turns out that they can be obtained from the independent model $Q(\mathbf{s}_t | \boldsymbol{\Theta}_t)$ that minimizes the following Kullback-Leibler (KL) divergence from $P(\mathbf{s}_t)$:

$$D(P||Q) = \sum_{\mathbf{s}_t} P(\mathbf{s}_t) \log \frac{P(\mathbf{s}_t)}{Q(\mathbf{s}_t)}. \tag{8}$$

The independent model $Q(\mathbf{s}_t | \boldsymbol{\Theta}_t^*) (\equiv Q^*(\mathbf{s}_t))$ that minimizes the KL divergence has activation rates $\mathbf{m}_t$ identical to those of the target distribution $P(\mathbf{s}_t)$[38] because the minimizing points $\Theta_{i,t}^*$ satisfy (for $i = 1, \ldots, N$)

$$\frac{\partial D(P||Q)}{\partial \Theta_{i,t}} \Big|_{\boldsymbol{\Theta}_t = \boldsymbol{\Theta}_t^*} = -\sum_{\mathbf{s}_t} s_{i,t} P(\mathbf{s}_t) + \tanh \Theta_{i,t}^*$$
$$= m_{i,t}^{Q^*} - m_{i,t}^P = 0, \tag{9}$$

where $m_{i,t}^P$ and $m_{i,t}^{Q^*}$ are respectively expectation values of $s_{i,t}$ by $P$

$(\mathbf{s}_t)$ and $Q(\mathbf{s}_t|\boldsymbol{\Theta}_t^*)$. As these values are equal, for the rest of the paper we will drop their superscripts and just write $m_{i,t}$ for simplicity. The result of this approximation is indifferent to the system's correlations. Later in the paper we will consider approximations that take into account pairwise correlations.

From an information geometric point of view, given $\mathbf{m}_t$ (or $\boldsymbol{\Theta}_t^*$), we may consider a family of points defined as a linear mixture of $P(\mathbf{s}_t)$ and $Q(\mathbf{s}_t|\boldsymbol{\Theta}_t^*)$ for which $\mathbf{m}_t$ is kept constant (the dashed line $\mathcal{A}$ in Fig. 1). This is known as an m-geodesic, and it is orthogonal to the e-flat manifold $\mathcal{Q}_t$, constituting an m-projection to this manifold[37,47]. Thus, the previous search of $Q(\mathbf{s}_t|\boldsymbol{\Theta}_t^*)$ given $P(\mathbf{s}_t|\mathbf{H},\mathbf{J})$ is equivalent to finding the orthogonal projection point from $P(\mathbf{s}_t|\mathbf{H},\mathbf{J})$ to the manifold $\mathcal{Q}_t$ of independent models[36,37].

**The Plefka expansion.** Although the m-projection provides the exact and unique average activation rates, its calculation in practice requires the complete distribution $P(\mathbf{s}_t)$. In the Plefka expansion, we relax the constraints of the m-projection, and introduce another set of more tractable distributions that passes only through $P(\mathbf{s}_t|\mathbf{H},\mathbf{J})$ and $Q(\mathbf{s}_t|\boldsymbol{\Theta}_t^*)$ (the solid line in Fig. 1). This distribution is defined using a new conditional distribution introducing a parameter $\alpha$ that connects a distribution on the manifold $\mathcal{Q}_t$ with the original distribution $P(\mathbf{s}_t)$:

$$P_\alpha(s_{i,t}|\mathbf{s}_{t-1}) = \frac{e^{s_{i,t}h_{i,t}(\alpha)}}{2\cosh h_{i,t}(\alpha)}, \quad (10)$$

$$h_{i,t}(\alpha) = (1-\alpha)\Theta_{i,t} + \alpha\left(H_i + \sum_j J_{ij}s_{j,t-1}\right). \quad (11)$$

At $\alpha=0$, $P_{\alpha=0}(\mathbf{s}_t|\mathbf{s}_{t-1}) = Q(\mathbf{s}_t|\boldsymbol{\Theta}_t)$, and $\alpha=1$ leads to $P_{\alpha=1}(\mathbf{s}_t|\mathbf{s}_{t-1}) = P(\mathbf{s}_t|\mathbf{s}_{t-1})$. Using this alternative conditional distribution $P_\alpha(s_{i,t}|\mathbf{s}_{t-1})$, we construct an approximate marginal distribution $P_\alpha(\mathbf{s}_t)$. Consequently, expectation values with respect to $P_\alpha(\mathbf{s}_t)$ are functions of $\alpha$. We thus write the statistics of the approximate system as $\mathbf{m}_t(\alpha)$, $\mathbf{C}_t(\alpha)$, and $\mathbf{D}_t(\alpha)$.

The Plefka expansions of these statistics are defined as the Taylor series expansion of these functions around $\alpha=0$. In the case of the mean activation rate, the expansion up to the $n$th-order leads to:

$$\mathbf{m}_t(\alpha) = \mathbf{m}_t(\alpha=0) + \sum_{k=1}^n \frac{\alpha^k}{k!}\frac{\partial^k \mathbf{m}_t(\alpha=0)}{\partial\alpha^k} + \mathcal{O}(\alpha^{(n+1)}), \quad (12)$$

where $\mathcal{O}(\alpha^{(n+1)})$ stands for the residual error of the approximation of order $n+1$ and higher. For the $n$th-order approximation, we neglect the residual terms as $\mathcal{O}(\alpha^{(n+1)})|_{\alpha=1} \approx 0$. Note that all coefficients of expansion are functions of $\boldsymbol{\Theta}_t$. The mean-field approximation is computed by setting $\alpha=1$ and finding the value of $\boldsymbol{\Theta}_t^*$ that satisfies Eq. (12). Since the original marginal distribution is recovered at $\alpha=1$, the equality of Eq. (9) holds: $\mathbf{m}_t(\alpha=1) = \mathbf{m}_t(\alpha=0)$. Then, we have

$$\sum_{k=1}^n \frac{1}{k!}\frac{\partial^k \mathbf{m}_t(\alpha=0)}{\partial\alpha^k} = 0, \quad (13)$$

which should be solved with respect to the parameters $\boldsymbol{\Theta}_t$. Since we neglected the terms higher than the $n$-th order, the solution may not lead to the exact projection, $Q(\mathbf{s}_t|\boldsymbol{\Theta}_t^*)$. In this study, we investigate the first $(n=1)$ and second $(n=2)$ order approximations. Moreover we can apply the same expansion to approximate the correlations $\mathbf{C}_t$ and $\mathbf{D}_t$, using Eq. (10).

What is the difference between this approach and other mean-field methods? Conventionally, naive mean-field approximations are obtained by minimizing $D(Q\|P)$ as opposed to $D(P\|Q)$ (Eq.

(8))[36,49]. This approach is typically used in variational inference to construct a tractable approximate posterior in machine learning problems. Following the Bogolyubov inequality, minimizing this divergence is equivalent to minimizing the variational free energy. Geometrically, it comprises an e-projection of $P(\mathbf{s}_t|\mathbf{H},\mathbf{J})$ to the submanifold $\mathcal{Q}_t$, which does not result in $Q(\mathbf{s}_t|\boldsymbol{\Theta}_t^*)$. Namely, minimizing $D(Q\|P)$, as well as minimization of other $\alpha$-divergences except for $D(P\|Q)$, introduces a bias in the estimation of the mean-field approximation[36,37]. In contrast, if we consider the m-projection point that minimizes $D(P\|Q)$, we can approximate the exact value of $\mathbf{m}_t$ using Eq. (12) up to an arbitrary order.

In the subsequent sections we show that different approximations of the marginal distribution $P(\mathbf{s}_t)$ in Eq. (3) can be constructed by replacing $P(s_{i,\tau}|\mathbf{s}_{\tau-1})$ with $P_\alpha(s_{i,\tau}|\mathbf{s}_{\tau-1})$ for different pairs $i,\tau$ (here we will explore the cases of $\tau=t$ and $\tau=t-1$). More generally, we show in Supplementary Note 1 that this framework can be extended to a marginal path of arbitrary length $k$, $P(\mathbf{s}_{t-k+1},\ldots,\mathbf{s}_t)$. In addition, we are not restricted to manifolds of independent models. The independent model is adopted as a reference model to approximate the average activation rate, but one can also more accurately approximate correlations using this method. In this vein, we can extend our framework to use reference manifolds $\mathcal{Q}_{t-k+1:t}$ (of models $Q(\mathbf{s}_{t-k+1},\ldots,\mathbf{s}_t)$) that include interactions, e.g., pairwise couplings between elements at two different time points, to more accurately approximate the delayed correlations (see Supplementary Note 1). By systematically defining these reference distributions, we will provide a unified approach to derive Plefka approximations of $\mathbf{m}_t$, $\mathbf{C}_t$, and $\mathbf{D}_t$, including the one that utilizes a pairwise structure.

**Plefka[$t-1,t$]: expansion around independent models at times $t-1$ and $t$.** Before elaborating different mean-field approximations, we demonstrate our method by deriving the known results of the classical nMF and TAP approximations for the kinetic Ising model[38,39]. In order to derive these classical mean-field equations, we make a Plefka expansion around the points $\boldsymbol{\Theta}_t^*$ and $\boldsymbol{\Theta}_{t-1}^*$ that are, respectively, obtained by orthogonal projection to the independent manifolds $\mathcal{Q}_t$ and $\mathcal{Q}_{t-1}$, computed as in Eq. (9). Here we should note that assuming an approximation where previous distributions (e.g., $t-2$, $t-3$, ...) are also independent yields exactly the same result. In this way, we derive the nMF and TAP equations of a model defined by a marginal probability distribution $P_\alpha^{[t-1:t]}$. Using Eqs. (3) and (10), we write

$$P_\alpha^{[t-1:t]}(\mathbf{s}_t) = \sum_{\substack{\mathbf{s}_{t-1}\\\mathbf{s}_{t-2}}} P_\alpha(\mathbf{s}_t|\mathbf{s}_{t-1})P_\alpha(\mathbf{s}_{t-1}|\mathbf{s}_{t-2})P(\mathbf{s}_{t-2}), \quad (14)$$

where $P_{\alpha=0}^{[t-1:t]}(\mathbf{s}_t) = Q(\mathbf{s}_t)$ and the original distribution is recovered for $P_{\alpha=1}^{[t-1:t]}(\mathbf{s}_t) = P(\mathbf{s}_t)$.

Following Eq. (13), for the first order approximation we have $\frac{\partial m_{i,t}(\alpha=0)}{\partial\alpha} = 0$. Since the derivative of the first order moment is

$$\frac{\partial m_{i,t}(\alpha=0)}{\partial\alpha} = (1-m_{i,t}^2)\left(-\Theta_{i,t} + H_i + \sum_j J_{ij}m_{j,t-1}\right), \quad (15)$$

by solving the equation, we find $\Theta_{i,t}^* \approx H_i + \sum_j J_{ij}m_{j,t-1}$ that leads to the naive mean-field approximation:

$$m_{i,t} \approx \tanh\left[H_i + \sum_j J_{ij}m_{j,t-1}\right]. \quad (16)$$

We apply the same expansion to approximate the correlations, expanding $C_{ik,t}(\alpha)$ and $D_{il,t}(\alpha)$ around $\alpha=0$ up to the first order

using $\Theta_{i,t} = \Theta_{i,t}^*$. Then we obtain

$$C_{ik,t} \approx 0, \quad i \neq k, \tag{17}$$

$$D_{il,t} \approx J_{il}(1 - m_{i,t}^2)(1 - m_{l,t-1}^2). \tag{18}$$

Detailed calculations are presented in Supplementary Note 2.

To obtain the second-order approximation, we need to solve $\frac{\partial m_i(\alpha=0)}{\partial \alpha} + \frac{1}{2}\frac{\partial^2 m_i(\alpha=0)}{\partial \alpha^2} = 0$ from Eq. (13). Here the second-order derivative is given as

$$\frac{\partial^2 m_{i,t}(\alpha=0)}{\partial \alpha^2} \approx -2m_{i,t}(1 - m_{i,t}^2)\sum_j J_{ij}^2(1 - m_{j,t-1}^2), \tag{19}$$

where terms of the order higher than quadratic were neglected (see Supplementary Note 2 for further details). From these equations, we find $\Theta_{i,t}^* \approx H_i + \sum_j J_{ij}m_{j,t-1} - m_{i,t}\sum_j J_{ij}^2(1 - m_{j,t-1}^2)$ leading to the TAP equation:

$$m_{i,t} \approx \tanh\left[H_i + \sum_j J_{ij}m_{j,t-1} - m_{i,t}\sum_j J_{ij}^2(1 - m_{j,t-1}^2)\right]. \tag{20}$$

Having $\Theta_{i,t}^*$, we can incorporate TAP approximations of the correlations by expanding $C_{ik,t}(\alpha)$ and $D_{il,t}(\alpha)$ (see Supplementary Note 2 for details) as:

$$C_{ik,t} \approx (1 - m_{i,t}^2)(1 - m_{k,t}^2)\sum_j J_{ij}J_{kj}(1 - m_{j,t-1}^2), \quad i \neq k, \tag{21}$$

$$D_{il,t} \approx J_{il}(1 - m_{i,t}^2)(1 - m_{l,t-1}^2)(1 + 2J_{il}m_{i,t}m_{l,t-1}). \tag{22}$$

In these approximations, Eqs. (16) and (20) of activation rates $\mathbf{m}_t$ correspond to the classical nMF and TAP equations of the kinetic Ising model[38,39]. The mean-field equations for the equal-time and delayed correlations (Eqs. (17), (18), (21), and (22)) are novel contributions from applying the Plefka expansion to correlations.

Using the equations above, we can compute the approximate statistical properties of the system at $t$ ($\mathbf{m}_t, \mathbf{C}_t, \mathbf{D}_t$) from $\mathbf{m}_{t-1}$. Therefore, the system evolution is described by recursively computing $\mathbf{m}_t$ from an initial state $\mathbf{m}_0$ (for both transient and stationary dynamics), although approximation errors accumulate over the iterations. After we introduce a unified view of mean-field approximations in the subsequent sections, we will numerically examine approximation errors of these various methods in predicting statistical structure of the system.

**Generalization of mean-field approximations.** In the previous section, we described a Plefka expansion that uses a model containing independent units at time $t-1$ and $t$ to construct a marginal probability distribution $P_\alpha^{[t-1:t]}(\mathbf{s}_t)$. This is, however, not the only possible choice of approximation. As we mentioned above, other approximations have been introduced in the literature. In ref. [40], expressions are provided for the nonstationary delayed correlations $\mathbf{D}_t$ as a function of $\mathbf{C}_{t-1}$. In ref. [41], an approximation is derived by assuming that units at state $\mathbf{s}_{t-1}$ are independent while correlations of $\mathbf{s}_t$ are preserved.

In the following sections, we show that various approximation methods, including those mentioned above, can be unified as Plefka expansions. Each method of the approximation corresponds to a specific choice of the submanifold $\mathcal{Q}_t$ at each time step. Fig. 2 shows the corresponding submanifolds $\mathcal{Q}_{t-1:t}$ of possible approximations, where gray lines represent interactions that are affected by $\alpha$ in the Plefka expansion. The mean-field approximations in the previous section were obtained by using the model represented in Fig. 2B, where the couplings at time $t-1$ and $t$ are affected by $\alpha$. Below, we present systematic applications of the Plefka expansions around other reference

models in order to approximate the original distribution (Fig. 2C–E). By doing so, we not only unify the previously reported mean-field approximations but also provide novel solutions that can provide more precise approximations than known methods.

**Plefka[$t$]: expansion around an independent model at time $t$.** For the Plefka[$t-1, t$] approximation, explained above, the system becomes independent for $\alpha = 0$ at $t$ as well as $t-1$. This leads to approximations of $\mathbf{m}_t, \mathbf{C}_t, \mathbf{D}_t$ being specified by $\mathbf{m}_{t-1}$, while being independent of $\mathbf{C}_{t-1}$ and $\mathbf{D}_{t-1}$. In ref. [40], the authors describe a mean-field approximation by performing new expansion over the classical nMF and TAP equations that takes into account previous correlations $\mathbf{C}_{t-1}$. Here, our framework allows us to obtain similar results by considering only a Plefka expansion over manifold $\mathcal{Q}_t$ while assuming that we know the properties of $P(\mathbf{s}_{t-1})$ (Fig. 2C). Therefore, we denote this approximation as $P_\alpha^{[t]}$ and consider

$$P_\alpha^{[t]}(\mathbf{s}_t) = \sum_{\mathbf{s}_{t-1}} P_\alpha(\mathbf{s}_t|\mathbf{s}_{t-1})P(\mathbf{s}_{t-1}). \tag{23}$$

In Supplementary Note 3 we derive the equations for this approximation. For the first order, we obtain

$$m_{i,t} \approx \tanh\left[H_i + \sum_j J_{ij}m_{j,t-1}\right], \tag{24}$$

$$C_{ik,t} \approx 0, \quad i \neq k, \tag{25}$$

$$D_{il,t} \approx (1 - m_{i,t}^2)\sum_j J_{ij}C_{jl,t-1}. \tag{26}$$

Note that Eqs. (24) and (25) are the same as the nMF Plefka[$t-1, t$] equations. Equation (26) includes $\mathbf{C}_{t-1}$, being exactly the same result obtained in ref. [40], Eq. (4). The second-order approximations leads to:

$$m_{i,t} \approx \tanh\left[H_i + \sum_j J_{ij}m_{j,t-1} - m_{i,t}\sum_{jl} J_{ij}J_{il}C_{jl,t-1}\right], \tag{27}$$

$$C_{ik,t} \approx (1 - m_{i,t}^2)(1 - m_{k,t}^2)\sum_{jl} J_{ij}J_{kl}C_{jl,t-1}, \quad i \neq k, \tag{28}$$

$$D_{il,t} \approx (1 - m_{i,t}^2)\sum_j J_{ij}C_{jl,t-1}(1 + 2J_{il}m_{i,t}m_{l,t-1}). \tag{29}$$

All update rules include the effect of $\mathbf{C}_{t-1}$. We can see that if we use the covariance matrix of the independent model at $t-1$, we recover the results of the Plefka[$t-1, t$] approximation in the previous section. In contrast with ref. [40], we provide a novel approximation method that depends on previous correlations using a single expansion (instead of two subsequent expansions), and additionally present approximated equal-time correlations.

**Plefka[$t-1$]: expansion around an independent model at time $t-1$.** In ref. [41], a mean-field method is proposed by approximating the effective field $\mathbf{h}_t$ as the sum of a large number of independent spins, approximated by a Gaussian distribution, yielding exact results for fully asymmetric networks in the thermodynamic limit. In our framework, we describe this approximation as an expansion around the projection point from $P(\mathbf{s}_{t-1})$ to the submanifold $\mathcal{Q}_{t-1}$, using a model where only $\mathbf{s}_{t-1}$ are independent (Fig. 2D). In this case (see Supplementary Note 4), the effective field $\mathbf{h}_t$ at the submanifold is a sum of independent terms, which for large $N$ yields a Gaussian distribution.

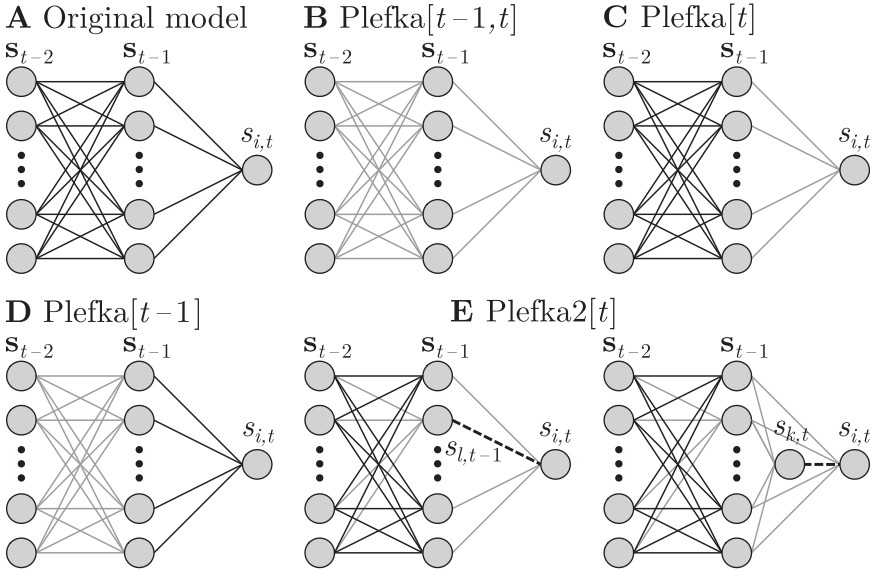

**Fig. 2 Unified mean-field framework.** Original model (**A**) and family of generalized Plefka expansions (**B–E**). Gray lines represent connections that are proportional to $\alpha$ and thus removed in the approximated model to perform the Plefka expansions, while solid black lines are conserved and dashed lines are free parameters. Plefka[$t-1, t$] (**B**) retrieves the classical naive and TAP mean-field equations[38,39]. Plefka[$t$] (**C**) results in a novel method which preserves correlations of the system at $t-1$, incorporating equations similar to ref. [40]. Plefka[$t-1$] (**D**) assumes independent activity at t-1, and in its first order approximation reproduces the results in ref. [41]. Plefka2[$t$] (**E**) represents a novel pairwise approximation which performs better in approximating correlations.

By defining

$$P_\alpha^{[t-1]}(\mathbf{s}_t, \mathbf{s}_{t-1}) = \sum_{\mathbf{s}_{t-2}} P(\mathbf{s}_t|\mathbf{s}_{t-1})P_\alpha(\mathbf{s}_{t-1}|\mathbf{s}_{t-2})P(\mathbf{s}_{t-2}), \qquad (30)$$

we see that now the expansion is defined for the marginal distribution of the path $\{\mathbf{s}_{t-1}, \mathbf{s}_t\}$ (see Supplementary Note 1). The first order equations for this method are

$$m_{i,t} \approx \int \mathrm{D}_x \tanh\left[H_i + \sum_j J_{ij}m_{j,t-1} + x\sqrt{\Delta_{i,t}}\right], \qquad (31)$$

$$C_{ik,t} \approx \int \mathrm{D}_{xy}^{\rho_{ik}} \tanh\left[H_i + \sum_j J_{ij}m_{j,t-1} + x\sqrt{\Delta_{i,t}}\right]$$
$$\cdot \tanh\left[H_k + \sum_l J_{kl}m_{l,t-1} + y\sqrt{\Delta_{j,t}}\right] - m_{i,t}m_{k,t}, \quad i \neq k, \qquad (32)$$

$$D_{il,t} \approx \sum_j J_{ij}C_{jl,t-1} \int \mathrm{D}_x \left(1 - \tanh^2\left[H_i + \sum_j J_{ij}m_{j,t-1} + x\sqrt{\Delta_{i,t}}\right]\right). \qquad (33)$$

Here we use $\mathrm{D}_x = \frac{dx}{\sqrt{2\pi}}\exp(-\frac{1}{2}x^2)$, $\mathrm{D}_{xy}^{\rho_{ik}} = \frac{dxdy}{2\pi\sqrt{1-\rho_{ik}^2}}$ $\exp(-\frac{1}{2}\frac{(x^2+y^2)-2\rho_{ik}xy}{1-\rho_{ik}^2})$, $\Delta_{i,t} = \sum_j J_{ij}^2(1-m_{j,t-1}^2)$ and $\rho_{ik} = \sum_j J_{ij}J_{kj}$ $(1-m_{j,t-1}^2)/\sqrt{\Delta_{i,t}\Delta_{j,t}}$. Derivations are described in Supplementary Note 4. These results are exactly the same as those presented for $\mathbf{m}_t, \mathbf{D}_t$ in ref. [41], adding an additional expression for $\mathbf{C}_t$. For this approximation, we do not consider the second-order equations since they are computationally much more expensive than the other approximations.

**Plefka2[$t$]: expansion around a pairwise model.** The proposed framework is also a powerful tool to develop novel Plefka expansions. To make the expansions more accurately

approximate target statistics, we can consider a reference manifold composed of multiple time steps while maintaining some of the parameters in the system (see Supplementary Note 1). Motivated by this idea, here we propose new methods that directly approximate pairwise activities of the units by choosing a reference manifold that preserves a coupling term.

Let us first consider the joint probability of any arbitrary pair of units at time $t-1$ and $t$ to compute the delayed correlations (Fig. 2E, left). Namely, we consider the joint probability of spins $s_{i,t}$ and $s_{l,t-1}$:

$$P(s_{i,t}, s_{l,t-1}) = \sum_{\substack{\mathbf{s}_{\backslash l,t-1} \\ \mathbf{s}_{t-2}}} P(s_{i,t}|\mathbf{s}_{t-1})P(\mathbf{s}_{t-1}|\mathbf{s}_{t-2})P(\mathbf{s}_{t-2}), \qquad (34)$$

with $\mathbf{s}_{\backslash l, t-1}$ containing all elements of $\mathbf{s}_{t-1}$ except $s_{l,t-1}$. As a reference manifold $\mathcal{Q}_{t-1:t}$, we consider the dependency among only the units $i$ and $l$:

$$Q(s_{i,t}, s_{l,t-1}) = Q(s_{i,t}|s_{l,t-1})Q(s_{l,t-1})$$
$$= \frac{e^{s_{i,t}\theta_{i,t}(s_{l,t-1})}}{2\cosh\theta_{i,t}(s_{l,t-1})}\frac{e^{s_{l,t-1}\Theta_{l,t-1}}}{2\cosh\Theta_{l,t-1}}, \qquad (35)$$

where $\theta_{i,t}(s_{l,t-1}) = \Theta_{i,t} + \Delta_{il,t}s_{l,t-1}$. The orthogonal projection to $\mathcal{Q}_t$ is equivalent to minimizing the KL divergence $D(P||Q)$ with respect to the parameters:

$$\left.\frac{\partial D(P||Q)}{\partial \Theta_{i,t}}\right|_{\substack{\theta_t = \theta_t^* \\ \Theta_{t-1} = \Theta_{t-1}^*}} = m_{i,t}^{Q^*} - m_{i,t}^P = 0, \qquad (36)$$

$$\left.\frac{\partial D(P||Q)}{\partial \Theta_{l,t-1}}\right|_{\substack{\theta_t = \theta_t^* \\ \Theta_{t-1} = \Theta_{t-1}^*}} = m_{l,t-1}^{Q^*} - m_{l,t-1}^P = 0, \qquad (37)$$

$$\left.\frac{\partial D(P||Q)}{\partial \Delta_{il,t}}\right|_{\substack{\theta_t = \theta_t^* \\ \Theta_{t-1} = \Theta_{t-1}^*}} = \langle s_{i,t}s_{l,t-1}\rangle_{(Q^*\cdot P)} - \langle s_{i,t}s_{l,t-1}\rangle_P = 0. \qquad (38)$$

with

$$\langle x \rangle_{(Q^* \cdot P)} = \sum_{\substack{s_{i,t} \\ s_{l,t-1}}} x \, Q(s_{i,t}|s_{l,t-1}, \boldsymbol{\theta}_t^*, \boldsymbol{\Theta}_{t-1}^*) P(s_{l,t-1}). \quad (39)$$

As in the previous approximations, $P(s_{i,t}, s_{l,t-1})$ is connected to $Q(s_{i,t}, s_{l,t-1}|\boldsymbol{\theta}_t^*, \boldsymbol{\Theta}_{t-1}^*)$ through an $\alpha$-dependent probability

$$P_\alpha^{2[t]}(s_{i,t}, s_{l,t-1}) = \sum_{\substack{\mathbf{s}_{\backslash l,t-1} \\ \mathbf{s}_{t-2}}} P_\alpha(s_{i,t}|\mathbf{s}_{t-1}) P_\alpha(s_{l,t-1}|\mathbf{s}_{t-2}) \\ \cdot P(\mathbf{s}_{\backslash l,t-1}|\mathbf{s}_{t-2}) P(\mathbf{s}_{t-2}), \quad (40)$$

with conditional probabilities given by

$$P_\alpha(s_{i,t}|\mathbf{s}_{t-1}) = \frac{e^{s_{i,t}h_{i,t}(\alpha)}}{2\cosh h_{i,t}(\alpha)},$$
$$h_{i,t}(\alpha) = (1-\alpha)\theta_{i,t}(s_{l,t-1}) + \alpha\left(H_i + \sum_j J_{ij}s_{j,t-1}\right), \quad (41)$$

$$P_\alpha(s_{l,t-1}|\mathbf{s}_{t-2}) = \frac{e^{s_{l,t-1}h_{l,t-1}(\alpha)}}{2\cosh h_{l,t-1}(\alpha)},$$
$$h_{l,t-1}(\alpha) = (1-\alpha)\Theta_{l,t-1} + \alpha\left(H_l + \sum_n J_{ln}s_{n,t-2}\right). \quad (42)$$

As in the cases above, we can calculate the equations for the first and second-order approximations (see Supplementary Note 5). Here, for the second-order approximation (which is more accurate than the first order) we have that:

$$\theta_{i,t}^*(s_{l,t-1}) \approx H_i + \sum_j J_{ij}m_{j,t-1} + J_{il}(s_{l,t-1} - m_{l,t-1})$$
$$+ \left(\sum_{j \neq l, n} J_{ij}J_{ln}D_{jn,t-1}\right)(s_{l,t-1} - m_{l,t-1}) \quad (43)$$
$$- \tanh\theta_{i,t}^*(s_{l,t-1}) \sum_{jn \neq l} J_{ij}J_{in}C_{jn,t-1},$$

$$\Theta_{l,t-1}^* \approx H_l + \sum_n J_{ln}m_{n,t-2} - m_{l,t-1}\sum_{mn} J_{lm}J_{ln}C_{mn,t-2}, \quad (44)$$

which directly leads to calculation of means and delayed correlations as:

$$m_{i,t} \approx \sum_{s_{l,t-1}} \tanh\theta_{i,t}^*(s_{l,t-1})Q^*(s_{l,t-1}), \quad (45)$$

$$m_{l,t-1} \approx \tanh\Theta_{l,t-1}^*, \quad (46)$$

$$D_{il,t} \approx \sum_{s_{l,t-1}} \tanh\theta_{i,t}^*(s_{l,t-1})s_{l,t-1}Q^*(s_{l,t-1}) - m_{i,t}m_{l,t-1}. \quad (47)$$

These results are related to previous work[43] that included autocorrelations as one of the constraints to derive the Plefka approximation. Instead, here we provide a Plefka approximation that includes delayed correlations between any pair of units.

To compute the above approximations, we need to know $\mathbf{C}_{t-1}$ and $\mathbf{C}_{t-2}$. Here, we provide similar pairwise Plefka approximations for the pairwise distribution at time $t$, $P(s_{i,t}, s_{k,t})$. Since $s_{i,t}, s_{k,t}$ are conditionally independent, we can construct a model in which first $s_{k,t}$ is computed from $\mathbf{s}_{t-1}$, and then $s_{i,t}$ is computed conditioned on $s_{k,t}, \mathbf{s}_{t-1}$ (Fig. 2E, right):

$$Q(s_{i,t}, s_{k,t}) = Q(s_{i,t}|s_{k,t})Q(s_{k,t}), \quad (48)$$

$$P_\alpha^{2[t]}(s_{i,t}, s_{k,t}) = \sum_{\mathbf{s}_{t-1}} P_\alpha(s_{i,t}|s_{k,t}, \mathbf{s}_{t-1})P_\alpha(s_{k,t}|\mathbf{s}_{t-1})P(\mathbf{s}_{t-1}), \quad (49)$$

with conditional probabilities given by

$$P_\alpha(s_{i,t}|s_{k,t}, \mathbf{s}_{t-1}) = \frac{e^{s_{i,t}h_{i,t}(\alpha)}}{2\cosh h_{i,t}(\alpha)},$$
$$h_{i,t}(\alpha) = (1-\alpha)\theta_{i,t}(s_{k,t})$$
$$+ \alpha\left(H_i + \sum_j J_{ij}s_{j,t-1}\right), \quad (50)$$

$$P_\alpha(s_{k,t}|\mathbf{s}_{t-1}) = \frac{e^{s_{k,t}h_{k,t}(\alpha)}}{2\cosh h_{k,t}(\alpha)},$$
$$h_{k,t}(\alpha) = (1-\alpha)\Theta_{k,t} + \alpha\left(H_k + \sum_l J_{kl}s_{l,t-1}\right). \quad (51)$$

Here $\theta_{i,t}$ is a function of $s_{k,t}$ that accounts for equal-time correlations between $s_{i,t}$ and $s_{k,t}$. Computed similarly to delayed correlations, the second-order approximation yields (see Supplementary Note 5):

$$\theta_{i,t}^*(s_{k,t}) \approx H_i + \sum_j J_{ij}m_{j,t-1} + \left(\sum_{jl} J_{ij}J_{kl}C_{jl,t-1}\right)(s_{k,t} - m_{k,t})$$
$$- \tanh\theta_{i,t}^*(s_{k,t})\sum_{jl} J_{ij}J_{il}C_{jl,t-1}, \quad (52)$$

$$\Theta_{k,t}^* \approx H_k + \sum_l J_{kl}m_{l,t-1} - m_{k,t}\sum_{jl} J_{kj}J_{kl}C_{jl,t-1}. \quad (53)$$

Using these equations, approximate equal-time correlations are given as

$$C_{ik,t} \approx \sum_{s_{k,t}} \tanh\theta_{i,t}^*(s_{k,t})s_{k,t}Q^*(s_{k,t}) - m_{i,t}m_{k,t}. \quad (54)$$

Note that the approximation of equal-time correlations may not be symmetric for $C_{ik,t}$ and $C_{ki,t}$. In the results of this paper we use the average of the two.

**Comparison of the different approximations**. In the subsequent sections, we compare the family of Plefka approximation methods described above by testing their performance in the forward and inverse Ising problems. More specifically, we compare the second-order approximations of Plefka$[t-1, t]$ and Plefka$[t]$, the first order approximation of Plefka$[t-1]$, and the second-order pairwise approximation of Plefka2$[t]$. We define an Ising model as an asymmetric version of the kinetic Sherrington-Kirkpatrick (SK) model, setting its parameters around the equivalent of a ferromagnetic phase transition in the equilibrium SK model. External fields $H_i$ are sampled from independent uniform distributions $\mathcal{U}(-\beta H_0, \beta H_0)$, $H_0 = 0.5$, whereas coupling terms $J_{ij}$ are sampled from independent Gaussian distributions $\mathcal{N}(\beta\frac{J_0}{N}, \beta^2\frac{J_\sigma^2}{N})$, $J_0 = 1$, $J_\sigma = 0.1$, where $\beta$ is a scaling parameter (i.e., an inverse temperature).

Generally, mean-field methods are suitable for approximating properties of systems with small fluctuations. However, there is evidence that many biological systems operate in critical, highly fluctuating regimes[5,6]. In order to examine different approximations in such a biologically plausible yet challenging situation, we select the model parameters around a phase transition point displaying large fluctuations.

To find such conditions, we employed path-integral methods to solve the asymmetric SK model (Supplementary Note 6). We find that the stationary solution of the asymmetric model displays for our choice of parameters a non-equilibrium analogue of a critical point for a ferromagnetic phase transition, which takes place at $\beta_c \approx 1.1108$ in thermodynamic limit (see Supplementary Note 6, Supplementary Fig. 1). The uniformly distributed bias

terms $\mathbf{H}$ shift the phase transition point from $\beta = 1$ obtained at $\mathbf{H} = \mathbf{0}$. By simulation of the finite size systems, we confirmed that the maximum fluctuations in the model are found near the theoretical $\beta_c$, which shows maximal covariance values (see Supplementary Note 6, Supplementary Fig. 2).

Fluctuations of a system are generally expected to be maximized at a critical phase transition[19]. In addition, entropy production (a signature of time irreversibility) has been suggested as an indicator of phase transitions. For example, it presents a peak at the transition point of a continuous phase transition in a non-equilibrium Curie-Weiss Ising model with oscillatory field[50] and some instances of mean-field majority vote models[51,52]. We found that the entropy production of the kinetic Ising system is also maximized around $\beta_c$ (discussed later, see also Methods for its derivation).

**Forward Ising problem**. We examine the performance of the different Plefka expansions in predicting the evolution of an asymmetric SK model of size $N = 512$ with random $\mathbf{H}$ and $\mathbf{J}$. To study the nonstationary transient dynamics of the model, we start from $\mathbf{s}_0 = \mathbf{1}$ (all elements set to 1 at $t = 0$) and recursively update its state for $T = 128$ steps. We repeated this stochastic simulation for $R = 10^6$ trials for 21 values of $\beta$ in the range $[0.7\beta_c, 1.3\beta_c]$ (except for the reconstruction of the phase transition where we used $R = 10^5$ and 201 values of $\beta$ in the same range). Using the $R$ samples, we computed the statistical moments and cumulants of the system, $\mathbf{m}_t$, $\mathbf{C}_t$, and $\mathbf{D}_t$ at each time step. We then computed their averages over the system units, i.e., $\langle m_{i,t} \rangle_i$, $\langle C_{ik,t} \rangle_{ik}$ and $\langle D_{il,t} \rangle_{il}$, where the angle bracket denotes average over indices of its subscript.

The black solid lines in Fig. 3A–C display nonstationary dynamics of these averaged statistics from $t = 0, \ldots, 128$, simulated by the original model at $\beta = \beta_c$. In comparison, color lines display these statistics predicted by the family of Plefka approximations that are recursively computed using the obtained equations, starting from the initial state $\mathbf{m}_0 = \mathbf{1}$, $\mathbf{C}_0 = \mathbf{0}$ and $\mathbf{D}_0 = \mathbf{0}$. We observe that although the recursive application of all the approximation methods provides good predictions for the transient dynamics of the mean activation rates $\mathbf{m}_t$ until its convergence (Fig. 3A), the predictions using Plefka[$t$] and especially the proposed Plefka2[$t$] approximations are closer to the true dynamics than the others. Evolution of the mean equal-time and time-delayed correlations $\mathbf{C}_t$, $\mathbf{D}_t$ is precisely captured only by our new method Plefka2[$t$]. In contrast, Plefka[$t$] overestimates correlations while Plefka[$t-1$] and Plefka[$t-1$, $t$] underestimate correlations.

Performance of the methods in predicting individual activation rates and correlations are displayed in Fig. 3D–F by comparing vectors $\mathbf{m}_t$, $\mathbf{C}_t$ and $\mathbf{D}_t$ at the last time step ($t = 128$) of the original model ($o$ superscript) and those of the Plefka approximations ($p$ superscript). For activation rates $\mathbf{m}_t$, the proposed Plefka2[$t$] and Plefka[$t$] perform slightly better than the others (see also Fig. 3A). While being overestimated by Plefka[$t$], underestimated moderately by Plefka[$t-1$] and significantly by Plefka[$t-1$, $t$], equal-time and time-delayed correlations $\mathbf{C}_t$, $\mathbf{D}_t$ are best predicted by Plefka2[$t$] (Fig. 3E, F).

The above results are obtained at the critical $\beta = \beta_c$, intuitively the most challenging point for mean-field approximations. In order to further show that our novel approximation Plefka2[$t$] systematically outperforms the others in a wider parameter range, we repeated the analysis for different inverse temperatures $\beta$ (the same random parameters are applied for all $\beta$). Fig. 3G, H, I, respectively, show the averaged squared errors (averaged over time and units) of the activation rates $\epsilon_{\mathbf{m}}$, equal-time correlations $\epsilon_{\mathbf{C}}$ and delayed correlations $\epsilon_{\mathbf{D}}$ between the original model and approximations, averaged over units and time for 21 values of $\beta$

in the range $[0.7\beta_c, 1.3\beta_c]$. Fig. 3G–I shows that Plefka2[$t$] outperforms the other methods in computing $\mathbf{m}_t$, $\mathbf{C}_t$, $\mathbf{D}_t$ (with the exception of a certain region of $\beta > \beta_c$ in which Plefka[$t$] is slightly better), yielding consistently a low error bound for all values of $\beta$. Errors of these approximations are smaller when the system is away from $\beta_c$.

**Inverse Ising problem**. We apply the approximation methods to the inverse Ising problem by using the data generated above for the trajectory of $T = 128$ steps and $R = 10^6$ trials to infer the parameters of the model, $\mathbf{H}$ and $\mathbf{J}$. The model parameters are estimated by the Boltzmann learning method under the maximum likelihood principle: $\mathbf{H}$ and $\mathbf{J}$ are updated to minimize the differences between the average rates $\mathbf{m}_t$ or delayed correlations $\mathbf{D}_t$ of the original data and the model approximations, which can significantly reduce computational time (see Methods). While Boltzmann learning requires to compute the likelihood of every point in a trajectory and every trial ($RT$ calculations) each iteration, we can estimate the gradient at each iteration in a one-shot computation by applying the Plefka approximations (Methods). At $\beta = \beta_c$ (Fig. 4A, B), we observe that the classical Plefka[$t-1$, $t$] approximation adds significant offset values to the fields $\mathbf{H}$ and couplings $\mathbf{J}$. In contrast, Plefka[$t$], Plefka[$t-1$] and Plefka2[$t$] are all precise in estimating the values of $\mathbf{H}$ and $\mathbf{J}$.

Fig. 4C, D shows the mean squared error $\epsilon_{\mathbf{H}}$, $\epsilon_{\mathbf{J}}$ for bias terms and couplings between the original model and the inferred values for different $\beta$. In this case, errors are large in the estimation of $\mathbf{J}$ for Plefka[$t-1$, $t$]. In comparison, Plefka[$t$], Plefka[$t-1$] and Plefka2[$t$] work equally well even in the high fluctuation regime ($\beta \approx \beta_c$). Since the inverse Ising problem is solved by applying approximation one single time step (per iteration), it is not as challenging as the forward problem that can accumulate errors by recursively applying the approximations. Therefore, different approximations other than the classical mean-field Plefka[$t-1$, $t$] perform equally well in this case.

**Phase transition reconstruction**. We have shown how different methods perform in computing the behavior of the system (forward problem) and inferring the parameters of a given network from its activation data (inverse problem). Combining the two, we can ask how well the methods explored here can reconstruct the behavior of a system from data, potentially exploring behaviors under different conditions than the recorded data.

First, in Fig. 5A–C we examine how the different approximation methods approximate fluctuations (equal-time and time-delayed covariances) and the entropy production (see Methods) at $t = 128$ after solving the forward problem by recursively applying the approximations for the 128 steps. As we mentioned above, the asymmetric SK model explored here presents maximum fluctuations and maximum entropy production around $\beta = \beta_c$ (Supplementary Note 6, Supplementary Fig. 2). However, we see that Plefka[$t-1$, $t$] and Plefka[$t-1$] cannot reproduce the behavior of correlations $\mathbf{C}_t$ and $\mathbf{D}_t$ of the original SK model around the transition point. Plefka[$t$] and Plefka2[$t$] show much better performance in capturing the behavior of $\mathbf{C}_t$ and $\mathbf{D}_t$ in the phase transition, although Plefka[$t$] overestimates both correlations. Additionally, all the methods capture the phase transition in entropy production, though Plefka[$t$] overestimates its value around $\beta_c$ and Plefka2[$t$] is more precise than the other methods.

Next, we combine the forward and inverse Ising problem and try to reproduce the transition in the asymmetric SK model in the models inferred from the data. We first take the values of $\mathbf{H}$, $\mathbf{J}$ from solving the inverse problem from the data sampled at $\beta = \beta_c$, and next we solve again the forward problem with those estimated

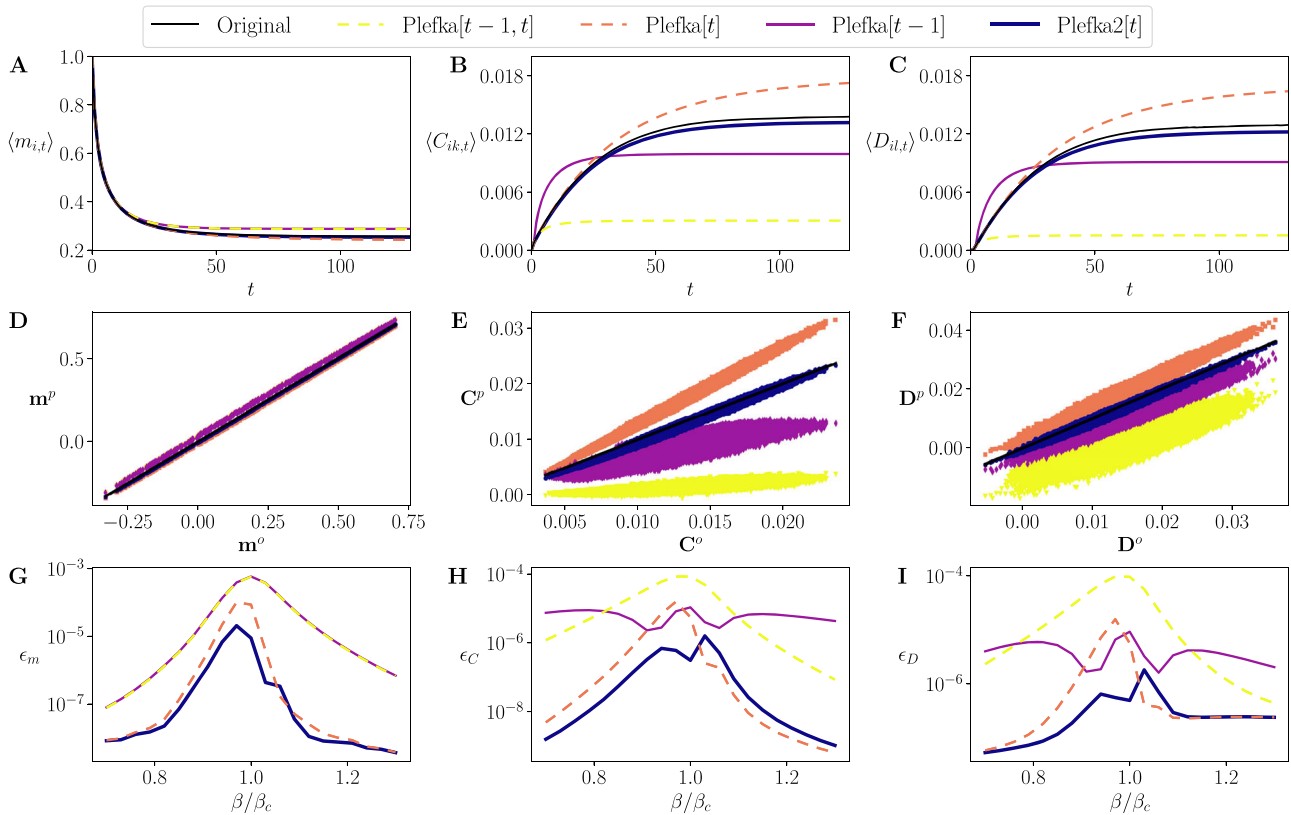

**Fig. 3 Forward Ising problem.** Top: Evolution of average activation rates (magnetizations) (**A**), equal-time correlations (**B**), and delayed correlations (**C**) found by different mean-field methods for $\beta = \beta_c$. Middle: Comparison of the activation rates (**D**), equal-time correlations (**E**), and delayed correlations (**F**) found by the different Plefka approximations (ordinate, $p$ superscript) with the original values (abscissa, $o$ superscript) for $\beta = \beta_c$ and $t = 128$. Black lines represent the identity line. Bottom: Average squared error of the magnetizations $\epsilon_{\mathbf{m}} = \langle\langle (m_{i,t}^o - m_{i,t}^p)^2 \rangle_i \rangle_t$ (**G**), equal-time correlations $\epsilon_{\mathbf{C}} = \langle\langle (C_{ik,t}^o - C_{ik,t}^p)^2 \rangle_{ik} \rangle_t$ (**H**), and delayed correlations $\epsilon_{\mathbf{D}} = \langle\langle (D_{il,t}^o - D_{il,t}^p)^2 \rangle_{il} \rangle_t$ (**I**) for 21 values of $\beta$ in the range $[0.7\beta_c, 1.3\beta_c]$.

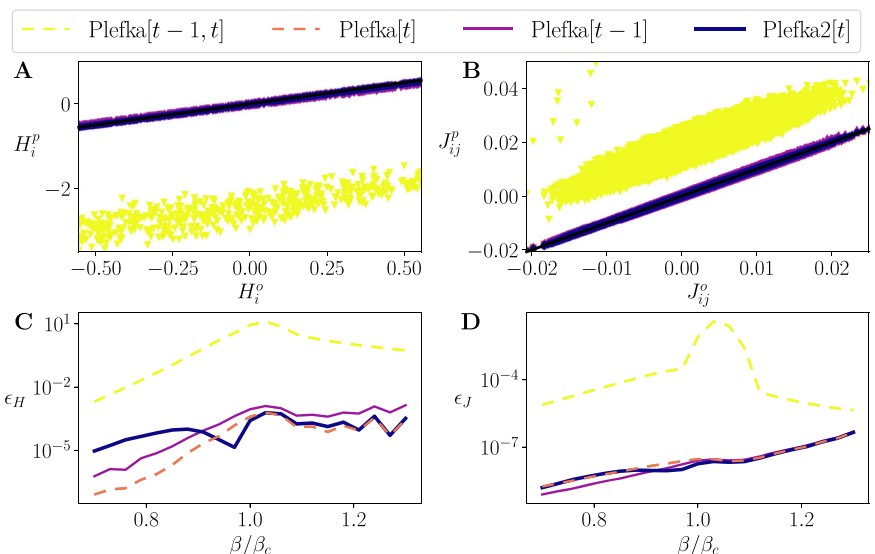

**Fig. 4 Inverse Ising problem.** Top: Inferred external fields (**A**) and couplings (**B**) found by different mean-field models plotted versus the real ones for $\beta = \beta_c$. Black lines represent the identity line. Bottom: Average squared error of inferred external fields $\epsilon_{\mathbf{H}} = \langle (H_i^o - H_i^p)^2 \rangle_i$ (**C**) and couplings $\epsilon_{\mathbf{J}} = \langle (J_{ij}^o - J_{ij}^p)^2 \rangle_{ij}$ (**D**) for 21 values of $\beta$ in the range $[0.7\beta_c, 1.3\beta_c]$.

parameters rescaled by a new inverse temperature $\tilde{\beta}$. The results for the correlations (Fig. 5D, E) show that in this case Plefka$[t-1,t]$ works badly, not being able to capture the transition. Plefka$[t-1]$ shows similar performances as in the forward problem, and Plefka$[t]$ and Plefka2$[t]$ have a similar behavior, underestimating

fluctuations slightly. When we analyze entropy production of the system (Fig. 5F), we find that Plefka2$[t]$ exhibits better performance with a high precision, with Plefka$[t-1]$ slightly overestimating it, Plefka$[t]$ underestimating it, and Plefka$[t-1,t]$ not capturing the phase transition. Overall, the results above suggest that Plefka2$[t]$ is

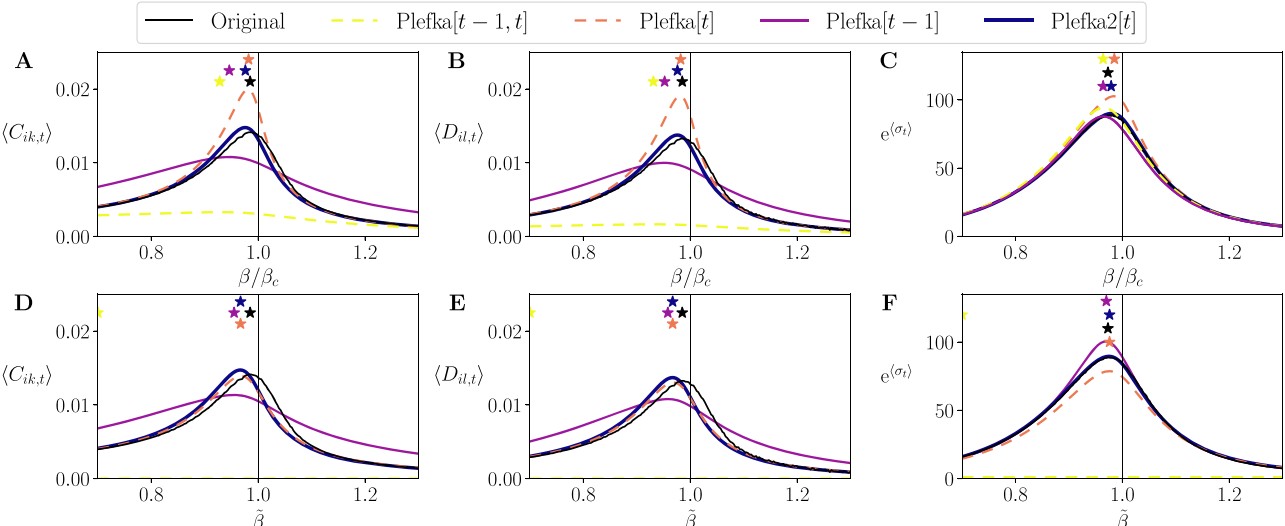

**Fig. 5 Reconstructing phase transition of kinetic Ising systems.** Top: Average of the Ising model's equal-time correlations (**A**), delayed correlations (**B**), and entropy production (shown as an exponential for better presentation of its maximum) (**C**), at the last step $t = 128$ found by different mean-field methods for $\beta = \beta_c$. Bottom (**D**–**F**): The same as above using the reconstructed network **H**, **J** by solving the inverse Ising problem at $\beta = \beta_c$ and multiplying a fictitious inverse temperature $\tilde{\beta}$ to the estimated parameters. The stars are marked at the values of $\tilde{\beta}$ that yield maximum fluctuations or maximum entropy production.

better suited to identify non-equilibrium phase transitions in models reconstructed from experimental data.

## Discussion

We have proposed a framework that unifies different mean-field approximations of the evolving statistical properties of non-equilibrium Ising models. This allows us to derive approximations premised on specific assumptions about the correlation structure of the system previously proposed in the literature. Furthermore, using our framework we derive a new approximation (Plefka2[$t$]) using atypical assumptions for mean-field methods, i.e., the maintenance of pairwise correlations in the system. This new pairwise approximation outperforms existing ones for approximating the behavior of an asymmetric SK model near the non-equilibrium equivalent of a ferromagnetic phase transition (see Supplementary Note 6), where classical mean-field approximations face problems. This shows that the proposed methods are useful tools to analyze large-scale, non-equilibrium dynamics near critical regimes expected for biological and social systems. However, we note that low-temperature spin phases (e.g., the spin-glass phase in symmetric models) also impose limitations on mean-field approximations[32,41], which could be further explored with methods like the ones presented here.

The generality of this framework allows us to picture other approximations with atypical assumptions. For example, the Sessak-Monasson expansion[53] for an equilibrium Ising model assumes a linear relation between $\alpha$ and spin correlations. An equivalent equilibrium expansion could use an effective field $h(\alpha)$ nonlinearly dependent on $\alpha$, satisfying linear $\mathbf{C}_t(\alpha) = \alpha \mathbf{C}_t$ or $\mathbf{D}_t(\alpha) = \alpha \mathbf{D}_t$ relations. As another extension, Plefka2[$t$] could incorporate higher-order interactions. As Eqs. (43) and (52) are each equivalent to two mean-field approximations with $s_{l,t-1} = \pm 1$ respectively, a generalized PlefkaM[$t$] would involve $2^{M-1}$ equations, increasing accuracy but also computational costs. In general, reference models $Q(\mathbf{s}_t)$ set coupling parameters of the model to zero at some steps of its dynamics. Other parameters (e.g., fields) are either free parameters fitted as m-projection from $P(\mathbf{s}_t)$, or preserved to their original value (see Supplementary Note 7 for comparing free and fixed parameters of each model).

Augmenting accuracy by increasing parameters often involves a computational cost. As a practical guideline for using each method, Supplementary Note 7 compares their precision and computation time in the forward and inverse problems (see also Supplementary Figs. 3 and 4).

Asides from its theoretical implications, our unified framework offers analysis tools for diverse data-driven research fields. In neuroscience, it has been popular to study the activity of ensembles of neurons by inferring an equilibrium Ising model with homogeneous (fixed) parameters[23] or inhomogeneous (time-dependent) parameters[25,54] from empirical data. Extended analyses based on the equilibrium model have reported that neurons operate near a critical regime[5,6]. However, studies of non-equilibrium dynamics in neural spike trains are scarce[7,26,55], partly due to the lack of systematic methods for analysing large-scale non-equilibrium data from neurons exhibiting large fluctuations. The proposed pairwise model Plefka2[$t$] is suitable for simulating such network activities, being more accurate than previous methods in predicting the network evolution at criticality (Fig. 3) and in testing if the system is near the maximally fluctuating regime (Fig. 5). In particular, application of our methods for computing entropy production in non-equilibrium systems could provide tools for characterizing the non-equilibrium dynamics of neural systems[56].

In summary, a unified framework of mean-field theories offers a systematic way to construct suitable mean-field methods in accordance with the statistical properties of the systems researchers wish to uncover. This is expected to foster a variety of tools to analyze large-scale non-equilibrium systems in physical, biological, and social systems.

## Methods

**Boltzmann learning in the inverse Ising problem.** Let $\mathbf{S}_t^r = \{S_{1,t}^r, S_{2,t}^r, \ldots, S_{N,t}^r\}$ for $t = 1, \ldots, T$ be observed states of a process described by Eq. (1) at the $r$-th trial ($r = 1, \ldots, R$). We also define $\mathbf{S}_{1:T}$ to represent the processes from all trials. The inverse Ising problem consists in inferring the external fields **H** and couplings **J** of the system. These parameters can be estimated by maximizing the log-likelihood $\ell(\mathbf{S}_{1:T})$ of the observed states under the model:

$$\ell(\mathbf{S}_{1:T}) = \log \prod_{t=1}^{T} \prod_{r=1}^{R} P(\mathbf{S}_t^r | \mathbf{S}_{t-1}^r) = \sum_t \sum_r \sum_i \left( S_{i,t}^r h_{i,t}^r - \log 2 \cosh h_{i,t}^r \right), \quad (55)$$

with $h^r_{i,t} = H_i + \sum_j J_{ij}S^r_{j,t-1}$. The learning steps are obtained as:

$$\frac{\partial \ell(\mathbf{S}_{1:T})}{\partial H_i} = RT(\langle S^r_{i,t}\rangle_{r,t} - \langle \tanh h^r_{i,t}\rangle_{r,t}), \tag{56}$$

$$\frac{\partial \ell(\mathbf{S}_{1:T})}{\partial J_{il}} = RT(\langle S^r_{i,t}S^r_{l,t-1}\rangle_{r,t} - \langle \tanh h^r_{i,t}S^{r,t}_{l,t-1}\rangle_{r,t}), \tag{57}$$

where $\langle\cdot\rangle_r$ denotes average over trials. We solve the inverse Ising problem by applying these equations as a gradient ascent rule adjusting $\mathbf{H}$ and $\mathbf{J}$. The second terms of Eqs. (56) and (57) need to be computed at every iteration, thus the computational cost grows linearly with $R \times T$. However, the use of mean-field approximations can significantly reduce the cost when a large number of samples $R$ and time bins $T$ are used to correctly estimate activation rates and correlations in large networks. Here the second terms can be written as

$$\langle \tanh h^r_{i,t}\rangle_{r,t} = \sum_{\mathbf{s},\tilde{\mathbf{s}}} s_i P(\mathbf{s}|\tilde{\mathbf{s}})\overline{P}(\tilde{\mathbf{s}}) = m_i, \tag{58}$$

$$\langle \tanh h^r_{i,t}S^r_{l,t-1}\rangle_{r,t} = \sum_{\mathbf{s},\tilde{\mathbf{s}}} s_i \tilde{s}_l P(\mathbf{s}|\tilde{\mathbf{s}})\overline{P}(\tilde{\mathbf{s}}) = D_{il} + m_i \tilde{m}_l, \tag{59}$$

where $\overline{P}(\tilde{\mathbf{s}}) = \frac{1}{RT}\sum_{r,t}\delta(\tilde{\mathbf{s}}, \mathbf{S}^r_t)$ is the empirical distribution averaged over trials and trajectories (with $\delta$ being a Kronecker delta) and $\tilde{m}_l$ is the average activation rate computed from the empirical distribution. $P(\mathbf{s}|\tilde{\mathbf{s}})$ is defined as Eq. (1). We then approximate $m_i$ and $D_{il}$ using the mean-field equations. Note that when we apply the mean-field equations, we replaced all statistics related to the previous step with those computed by the empirical distribution. By applying the mean-field methods, we reduced the computation of $R$ trials of trajectories of length $T$ into a single computation (instead of $RT$ calculations). In our numerical tests, gradient ascent was executed using learning coefficients $\eta_H = 0.1/RT, \eta_J = 1/(RT\sqrt{N})$, starting from $\mathbf{H} = \mathbf{0}, \mathbf{J} = \mathbf{0}$.

**Entropy production of the kinetic Ising model.** The entropy production is defined as the KL divergence between the forward and backward path, quantifying the irreversibility of the system[17,55,57]:

$$\langle \sigma_t \rangle = \frac{1}{2}\sum_{\mathbf{s}_t, \mathbf{s}_{t-1}} (P(\mathbf{s}_{t-1})P(\mathbf{s}_t|\mathbf{s}_{t-1}) - P(\mathbf{s}_t)P_B(\mathbf{s}_{t-1}|\mathbf{s}_t)) \log \frac{P(\mathbf{s}_t|\mathbf{s}_{t-1})P(\mathbf{s}_{t-1})}{P_B(\mathbf{s}_{t-1}|\mathbf{s}_t)P(\mathbf{s}_t)}. \tag{60}$$

where $P_B(\mathbf{s}_{t-1}|\mathbf{s}_t)$ is a probability of the backward trajectory defined as in Eq. (1) but switching $\mathbf{s}_t$ and $\mathbf{s}_{t-1}$. Assuming a non-equilibrium steady state, where $P(\mathbf{s}_t) = P(\mathbf{s}_{t-1})$, the entropy production of the kinetic Ising system is computed as:

$$
\begin{aligned}
\langle \sigma_t \rangle = \sum_{\mathbf{s}_t, \mathbf{s}_{t-1}} P(\mathbf{s}_t, \mathbf{s}_{t-1}) &\left( \sum_i H_i(s_{i,t} - s_{i,t-1}) + \sum_{ij} J_{ij}(s_{i,t}s_{j,t-1} - s_{i,t-1}s_{j,t}) \right.\\
&- \log\left(2\cosh\left(H_i + \sum_j J_{ij}s_{j,t-1}\right)\right) + \log\left(2\cosh\left(H_i + \sum_j J_{ij}s_{j,t}\right)\right) \Bigg)\\
&+ \log P(\mathbf{s}_{t-1}) - \log P(\mathbf{s}_t) = \sum_{ij} J_{ij}(D_{ij,t} - D_{ji,t}) = \sum_{ij}(J_{ij} - J_{ji})D_{ij,t}.
\end{aligned}
\tag{61}
$$

## Data availability

The datasets generated and analysed in this study are available under CC BY license at Zenodo https://zenodo.org/record/4318983[58] (https://doi.org/10.5281/zenodo.4318983).

## Code availability

The source code for implementing the methods and results in this work is available under GPL license at GitHub https://github.com/MiguelAguilera/kinetic-Plefka-expansions[59] (https://doi.org/10.5281/zenodo.4357634).

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

## Acknowledgements

We thank Yasser Roudi and Masanao Igarashi for valuable comments and discussions on this manuscript. This work was supported in part by the Cooperative Intelligence Joint Research between Kyoto University and Honda Research Institute Japan, MEXT/JSPS KAKENHI Grant Number JP 20K11709, and the grant of Joint Research by the National Institutes of Natural Sciences (NINS Program No. 01112005). M.A. was funded by the European Union's Horizon 2020 research and innovation programme under the Marie Skłodowska-Curie grant agreement No 892715 and the University of the Basque Country UPV/EHU post-doctoral training program grant ESPDOC17/17, and supported in part by the Basque Government project IT 1228-19  and project Outonomy PID2019-104576GB-I00 by the Spanish Ministry of Science and Innovation.

## Author contributions

M.A., S.A.M., and H.S. designed and reviewed research; M.A. contributed analytical and numerical results; M.A., S.A.M., and H.S. wrote the paper.

## Competing interests

The authors declare no competing interests.
