## [Peer Review File · Nature Communications]

REVIEWER COMMENTS

Reviewer #1 (Remarks to the Author):

The paper provided a unifying framework for various types of the Plefka expansion, from which mean field approximations can be derived. By utilizing the perspective of information geometry, the authors clearly explained the family of Plefka expansions, which was previously proposed, in a unified manner, and then, even proposed a novel Plefka expansion. They showed their novel method outperformed the previous methods in accurately approximating the covariances and in inferring the model parameters from data.

The framework provided in this paper will be very helpful for understanding relationship between different types of Plefka expansions. Not only that, the authors also successfully invented a novel better way of the Plefka expansion than the previous ones, which showed the utility and strength of the proposed framework. I think that the proposed unifying framework will provide a significant theoretical advancement in the field. On the other hand, when we consider the novel Plefka expansion, $\text{Plefka2}[t]$, as a data analysis tool, it is not that clear to me whether the advance is incremental or it would provide even "qualitatively" different results, which would not be obtained by the previous methods. To clarify and strengthen the significance of $\text{Plefka2}[t]$ in practical applications, the authors are encouraged to address the following comments below.

(1) Theoretical comparison between the novel method Plefka2 and other existing methods as a data analysis tool

I feel that the authors should provide further theoretical comparison between different Plefka expansions. First, I would like the authors to clarify the inclusion relation between the different probabilistic models used in the different Plefka expansions. From the schematic shown in Fig. 2, $\text{Plefka}[t-1, t]$ seems to be the simplest model and this model is included in the more complex models; $\text{Plefka}[t]$, $\text{Plefka}[t-1]$, and $\text{Plefka2}[t]$. Thus, I think we can expect without any simulation that $\text{Plefka}[t-1, t]$ gives the worst approximation if there is enough amount of data. On the other hand, $\text{Plefka2}[t]$ does not seem to include $\text{Plefka}[t-1]$ and $\text{Plefka}[t]$. In this sense, it is not entirely obvious that $\text{Plefka2}[t]$ gives the best approximation. However, since $\text{Plefka2}[t]$ seems to have the greatest number of model parameters, i.e., the most complex model, it is more likely that $\text{Plefka2}[t]$ gives the best approximation. I am not sure about the number of parameters used for estimating m and D in Eqs. (52) and (53) for each Plefka expansion, and the inclusion relation between the different Plefka expansions. Please clarify this point first and discuss whether we can naturally expected that $\text{Plefka2}[t]$ will give better approximations than $\text{Plefka}[t-1]$ and $\text{Plefka}[t]$.

(2) Comparison between Plefka2 and other possible Plefka expansions

Next, if we can expect that the more complex a model is, the more accurately it can estimate H and J in the inverse Ising problem, we would get a more accurate Plefka expansion by simply increasing the complexity of the model. For example, we could consider the second order Taylor expansion, or " $\text{Plefka3}[t]$ " where even triple-wise activities are considered as well as pair-wise activities. If the objective is to get the most accurate approximation, we could further increase the accuracy of approximations with such more complicated methods. I think that the superiority of $\text{Plefka2}[t]$, which the authors proposed, to other possible approximations should be discussed in the context of the trade-off between the model complexity and the computational complexity. In the paper, there are no discussions in this aspect, i.e., which model is the most efficient considering both the computational costs and the accuracy of estimates.

(3) Qualitative difference between Plefka2[t] and Plefka[t] & [t-1] in practical applications

The authors quantitatively showed in Fig. 4 that Plefka2 is the most accurate model for estimating H and J. I can of course understand Plefka2 provides the least errors compared with other methods, but I cannot intuitively tell how significant this reduction of errors is in practical applications. For example, as written in Discussion, Ising models have been used to tell whether a system operates near a critical regime. In this application, how much difference would Plefka2 make in estimating critical points compared with the case where other Plefka expansions are used?

Furthermore, the authors stated that Plefka[t] and Plefka[t-1] actually “does not converge”. I do not exactly understand what this “does not converge” mean but I interpret this as the fact that we cannot basically use Plefka[t] and Plefka[t-1] as a valid estimation method. Is that correct? If so, I think this is a huge qualitative difference between Plefka2[t] and Plefka[t] & [t-1] and the authors should emphasize this fact more to strengthen the advantage of Plefka2[t]. Interestingly, Plefka[t,t-1] is better than Plefka[t] & [t-1] although Plefka[t,t-1] is the simplest and the most crude approximation, which also probably means that there are some practical flaws in Plefka[t] & [t-1]. I would like the authors to clarify and discuss this point more.

Reviewer #2 (Remarks to the Author):

The paper introduces an unifying framework for mean field approximations to the dynamics of asymmetric kinetic Ising systems. While the paper is interesting and the contribution has potential, the writing has important flaws that make it hard to judge the content properly. Below, I provide a list of my main concerns, followed by a number of suggestions.

Main concerns

1. The paper seems to be written for a highly specialised audience, and hence unless a substantial changes are done it might not fit a generalist journal like Nature Communications. In particular, there is not enough argumentation of why non-specialists should care about the kinetic Ising model, mean field approximations, or the contribution proposed by the authors. Furthermore, the paper focuses on technicalities of the method but do not provide applications to real data that could provide evidence of its value for people who could want to use the model. As it stands, the paper fails to effectively speak to a larger audience — beyond people already invested in kinetic Ising models.
2. It is not clear which is exactly the contribution of the paper. It seems that some of these mean-field approximations existed beforehand and some others didn't, but this is not explicitly clarified. Also, how relevant is the asymmetry for the contribution? Please clarify all these issues, explaining why this is relevant for a generalist audience.
3. While the authors claim that their method is based on information geometry, I failed to see how exactly information geometry is driving their approach. From what I see in pg.3, P_α is just another 1-dim manifold that passes through P and Q, but is not clear if it is also a geodesic or some metric. Moreover, I don't see specific tools of information geometry at play in the derivation or application of the method. Given the importance that the authors place on information geometry when describing their framework, the authors should develop more this aspect of the work.

Suggestions

1. In the beginning of the introduction, the authors argue about the importance of non-equilibrium dynamics. However, it is not clear if they are really talking about non-equilibrium, in a thermodynamic system, or just talking about non-stationary. As I don't see any real thermodynamics at play through the paper (i.e. work, heat, etc), I suggest to modify the discussion and state it in terms of non-stationarity, this to avoid confusion and misinterpretations.
2. Please add some references when first introducing the kinetic Ising model in the introduction.
3. It would be appreciated to have a bit more of discussion about the limitations of direct fitting kinetic Ising models in real data. Is it just computationally expensive, or plainly intractable? Which kind of sizes are still workable? (considering that nowadays some companies are working with statistical models of hundreds of billions or parameters). This discussion is important, as it provides the reason of existence of the approximations that motivate this work.
4. When discussing symmetry and asymmetry in kinetic Ising models, it is not clear what kind of symmetry the authors are referring too. Please clarify this, and explain why this is important for theoretical and also practical applications.
5. In the section "The Kinetic Ising Model", the model is not clearly specified. For example, Eq. (1) specify the conditional probability of individual units, and then Eq. (3) uses the joint conditional probability of the whole system. How are these two related? Is the joint conditional equal to the product of the individual conditionals? Also, it is assumed that the temporal dynamics are Markovian (i.e. that the system's state at time t do not depend to past states $t-2$, $t-3$... given the state at $t-1$). Finally, shouldn't an initial condition be specified in order to have a well-defined stochastic process?
6. I don't see a need of mention Plefka expansions in pg.2.
7. When introducing the KL divergence in Eq. (8), it is not clear it is the relevant metric to be used. There are many non-negative divergences that quantify "distances" between two joint probabilities, so, why the KL is the relevant one here? Please provide a motivation there.
8. The Plefka approximation is mentioned many times before it is defined. Also, it feels awkward that something so important for the paper is defined within an example.
9. In the example provided in pg.4, it is not clear what the reader should focus in, or why it should care. Please provide more description of what is the purpose of the example.
10. The caption of Fig.2 fails to provide an clear explanation of the point of the figure, while calling a number of references. This might appeal an specialist but not to the general audience.
11. Figure 2 seems to have something odd with the labels, as the last figure has no letter (should it be F?), and the name in front of E is way away of it.

Reviewer #3 (Remarks to the Author):

The paper unifies and extends various existing mean-field approximations of the kinetic Ising model and presents in addition new approximations based on the Plefka expansion. The various approaches are carefully compared and discussed.

This work is to my knowledge novel, and I think that the manuscript is valid and worth of publication in NATURE Communications.

However, one issue could be clarified (or at least mentioned as an unsolved problem) by the authors:

The validity of the static TAP equations is restricted in the spin glass regime according to [22] . A specific, but important example for an incorrect TAP solution in the zero field situation is the paramagnetic solution with vanishing local magnetisations below the spin glass temperature.

Similar restrictions are expected for the dynamical case. The question is what is going on, if the dynamics enters the prohibited region of the statics. To the best of my knowledge, these problems have not been worked out for mean field approximations of the kinetic Ising systems

As a minor remark I recommend to enlarge Fig. 3 and Fig.4 . In the present form they are tiny.

With these amendments I thus recommend this work for publication.

General comments

We appreciate the reviewers' critical comments, which have allowed us to improve our manuscript significantly. Before responding to each review in detail, here we provide a summary of major changes in the Methods and Results section:

1. In responding to Reviewer 1, we found that our method $\text{Plefka2}[t]$ for approximating the equal-time correlations (C_t) can be improved by considering a more suitable reference distribution. Thus, we revised our $\text{Plefka2}[t]$ method for approximating the equal-time correlations (note that the $\text{Plefka2}[t]$ method for delayed correlations is unchanged). The revised approximation is much more accurate than the previous one in predicting the equal-time correlations. Note that we previously reported similar performance for $\text{Plefka2}[t]$ and $\text{Plefka}[t]$ in approximating C_t . In the current version we show that $\text{Plefka2}[t]$ not only outperforms the others in capturing dynamics of delayed correlations but also more accurately approximates the equal-time correlations.

2. To respond to Reviewer 1, we investigated our previous report on failure of $\text{Plefka}[t]$ and $\text{Plefka}[t-1]$ methods in the inverse Ising problem. We found that our approach to computing the gradient involved a numerical error when computing Eq. 53 in the previous manuscript, plus a small sample size (which we increased from $1E5$ to $1E6$ samples). These errors did not affect the forward problem, but affected the inverse problem as it interacts with data in a recursive manner. Fixing this issue resulted in improved performance of these methods. As a result, the studied methods, except for $\text{Plefka}[t-1,t]$, perform equally well. Thus we no longer claim the superiority of $\text{Plefka2}[t]$ for the inverse Ising problem. However, $\text{Plefka2}[t]$ performs better in the forward problem as well as in the newly introduced analysis on the phase transition (see below). We note that the goal of our work is to present a unifying framework for plefka expansions by which we can systematically propose new approximation methods. In this view $\text{Plefka2}[t]$ is a relevant example to show how we can go beyond the usual mean-field approximations.

3. In response to Reviewer 2, we tried to illustrate the application of our methods by reconstructing the phase transition found in the SK model by combining the forward and inverse Ising problems (a procedure that could be replicated in experimental data like neural recordings), showing how inference of parameters and simulation using these parameters can reproduce the fluctuations and entropy production in the model. We added a short subsection at the end of the Results section comparing the different methods in this task.

4. In order to fit the manuscript length to the journal's guidelines, we replaced the major parts of previous discussion with discussion that addresses the reviewer's comments. We also shortened the abstract to make it approximately 150 words.

Response to reviewers

Reviewer #1 (Remarks to the Author):

The paper provided a unifying framework for various types of the Plefka expansion, from which mean field approximations can be derived. By utilizing the perspective of information geometry, the authors clearly explained the family of Plefka expansions, which was previously proposed, in a unified manner, and then, even proposed a novel Plefka expansion. They showed their novel method outperformed the previous methods in accurately approximating the covariances and in inferring the model parameters from data.

The framework provided in this paper will be very helpful for understanding relationship between different types of Plefka expansions. Not only that, the authors also successfully invented a novel better way of the Plefka expansion than the previous ones, which showed the utility and strength of the proposed framework. I think that the proposed unifying framework will provide a significant theoretical advancement in the field. On the other hand, when we consider the novel Plefka expansion, $\text{Plefka2}[t]$, as a data analysis tool, it is not that clear to me whether the advance is incremental or it would provide even “qualitatively” different results, which would not be obtained by the previous methods. To clarify and strengthen the significance of $\text{Plefka2}[t]$ in practical applications, the authors are encouraged to address the following comments below.

We thank Reviewer 1 for appreciating the work and even encouraging us to strengthen the suggested method, Plefka2 . Below we provide our answers to the comments, and changes made to address the issues.

(1) Theoretical comparison between the novel method Plefka2 and other existing methods as a data analysis tool

I feel that the authors should provide further theoretical comparison between different Plefka expansions. First, I would like the authors to clarify the inclusion relation between the different probabilistic models used in the different Plefka expansions. From the schematic shown in Fig. 2, $\text{Plefka}[t-1, t]$ seems to be the simplest model and this model is included in the more complex models; $\text{Plefka}[t]$, $\text{Plefka}[t-1]$, and $\text{Plefka2}[t]$. Thus, I think we can expect without any simulation that $\text{Plefka}[t-1, t]$ gives the worst approximation if there is enough amount of data. On the other hand, $\text{Plefka2}[t]$ does not seem to include $\text{Plefka}[t-1]$ and $\text{Plefka}[t]$. In this sense, it is not entirely obvious that $\text{Plefka2}[t]$ gives the best approximation. However, since $\text{Plefka2}[t]$ seems to have the greatest number of model parameters, i.e., the most complex model, it is more likely that $\text{Plefka2}[t]$ gives the best approximation. I am not sure about the number of parameters used for estimating m and D in Eqs.

(52) and (53) for each Plefka expansion, and the inclusion relation between the different Plefka expansions. Please clarify this point first and discuss whether we can naturally expected that $\text{Plefka2}[t]$ will give better approximations than $\text{Plefka}[t-1]$ and $\text{Plefka}[t]$.

We thank the reviewer for pointing out this issue, and allowing us to clarify the advantages of our novel approximation, Plefka2[t]. As correctly noted by the reviewer, the reference model in Plefka[t-1,t] is a submodel of the other three reference models used in Plefka[t], Plefka[t-1], and Plefka2[t]. The the reference manifolds for the three approximation models (Plefka[t], Plefka[t-1], Plefka2[t]) can have mutual intersections but none of them can be considered as a subset of the other that is because each model manifests a distinct assumption on correlations. To clarify the relation between the performance of each approximation and its model complexity, we provide the number of free and fixed parameters of each model (see below). However, we think that in addition to the number of free parameters, the structure of the reference model is also critical in determining whether the approximation performs well or not. In the manuscript, we demonstrated that one could choose a suitable model depending on the statistics one wishes to approximate.

To explain these points, let us first revisit the definition of each approximated model, and introduce the number of free or fixed parameters in each model. Each approximation of P (a marginal distribution of spins at time t) defines a reference manifold using Q to perform a Plefka expansion. Q is defined at each time point, and obtained by fixing some parameters (couplings) of P at zero. The rest of the parameters are either free parameters that are fitted as a (m-)projection from P, or preserved as the original values of P. In the table below, we listed the number of free and fixed (zero or original) parameters of reference models. For example, in Plefka[t], we preserve one field parameter (h_i) and N couplings (J_{ij}) for each i-th neuron at time t-1 as the original value, while fixing N couplings of the i-th neuron at time t to zero. A field parameter of the i-th neuron at time t is a free parameter obtained as an m-projection from P (i.e., it is fitted so that Q and P have the expectation $m_{i,t}$). We need to perform this projection for all neurons (i.e., N times).

	Number of times the approximation is computed	Free parameters to be fitted (at each approximation)	Parameters fixed at zeros (at each approximation)	Parameters fixed at the original values (at each approximation)
Plefka[t-1,t]	2xN (N neurons at time t and N neurons at time t-1)	1 field	N couplings	0
Plefka[t]	N (neurons at time t)	1 field	N couplings (at time t)	1 field + N couplings (for N neurons at t-1)

Plefka[t-1]	N (neurons at time t-1)	1 field	N couplings (at time t-1)	1 field + N couplings (for N neurons at t)
Plefka2[t] (for approximating D_t)	$N \times N$ (once for every pair of neurons at time t)	2 fields (time t and t-1), 1 coupling	(N-1) couplings at t, N couplings at t-1	1 field + N couplings (for N-1 neurons except for “I” at t-1)

In general, to make expansions more accurate, we can make Q closer to P by fixing less original parameters at zeros, and making more parameters be fitted as a projection from P. When we compare Plefka[t-1,t] with Plafka[t] and Plefka[t-1], they share the same number of free parameters (1 field parameter for each spin). However Plefka[t] and Plefka[t-1] preserve the field and couplings at time t-1 and t, while Plefka[t-1,t] makes couplings at both time t and t-1 fixed at zeros. Therefore, Plefka[t-1,t] is the most distant from the original model P. When we compare Plefka2[t] with Plefka[t] and Plefka[t-1], the number of parameters fixed at original values in Plefka2[t] is slightly less than those of Plefka[t] and Plefka[t-1]. However, the number of free parameters of the reference model Q in Plefka2[t] is larger than those of Plefka[t] and Plefka[t-1], which can make the reference model of Plefka2[t] closer to P.

However, here we stress that choice of the reference model is crucial in addition to the number of free parameters. Namely, we could choose Q that preserves a coupling that plays a key role in approximating the target statistic. Plefka2[t] for approximating the delayed correlation (D_t) assumes the marginal pairwise model that preserves a coupling only for the target pair. By fitting the coupling (by m-projection), we can expect that Q shares the same joint rates of spins at different time-steps as for the original model. The method of Plefka expansion provides its approximation. Thus we expect that Plefka2[t] gives better approximations than Plefka[t-1] and Plefka[t] in approximating the delayed correlations D_t .

In responding to this question by the reviewer, we realized that Plefka2[t] for the equal-time correlations (C_t) does not preserve the key coupling for this statistics. **In the revised manuscript, we revised the method, and indeed this revised method provided significantly better approximation of C_t than the previous approach and the other methods (Please note that the previous Plefka2[t] performed as good as Plefka[t], therefore we did not claim its superiority in approximating C_t).**

The above argument provides why we can ‘naturally expect’ that Plefka2[t] can perform better than the others. We included these arguments in the revised manuscript as follows.

- 1. We introduced a brief discussion on how the number of free and fixed parameters, and the choice of reference manifold influences the approximation performance in the Discussion section.**

2. We included the above table in Supplementary Information (SI. 7).
3. We more clearly motivated the introduction of our Plefka2[t] that directly aims at estimating the target statistic.
4. We revised the method of Plefka2[t] in approximating the equal-time correlations, C_t .

(2) Comparison between Plefka2 and other possible Plefka expansions

Next, if we can expect that the more complex a model is, the more accurately it can estimate H and J in the inverse Ising problem, we would get a more accurate Plefka expansion by simply increasing the complexity of the model. For example, we could consider the second order Taylor expansion, or "Plefka3[t]" where even triple-wise activities are considered as well as pair-wise activities. If the objective is to get the most accurate approximation, we could further increase the accuracy of approximations with such more complicated methods. I think that the superiority of Plefka2[t], which the authors proposed, to other possible approximations should be discussed in the context of the trade-off between the model complexity and the computational complexity. In the paper, there are no discussions in this aspect, i.e., which model is the most efficient considering both the computational costs and the accuracy of estimates.

We agree with the reviewer that "the superiority of Plefka2[t] should be discussed in the context of the trade-off between the model complexity and the computational complexity." The difference between Plefka[t] and Plefka2[t] is that in Plefka[t] we expand around one independent manifold, but in the latter Plefka2[t], we independently expand around $N*(N-1)$ different pairwise submanifolds (one submanifold for each pair). Thus this accurate expansion comes with the cost of performing many projections.

In the revised manuscript, **we commented about the trade-off between model complexity and accuracy in the Discussion, and provided new supplementary information about the accuracy vs. computational cost (SI.7). There, we calculated the computation time for each method, and added it as a supplementary figure.** It will be a useful guideline to select the method for users of these methods, including our accurate method that comes with the cost of computational time.

Regarding the triple-wise interaction model, we believe that the intuition of the reviewer is correct. Although the kinetic Ising model is based on pairwise interactions, the marginal distribution (Eq.3) contains higher-order interactions. Therefore, such a reference model that includes the triple-wise interactions would provide better approximations, but with computational costs. **In the discussion we mention the possibility of obtaining higher order approximations and the compromise between accuracy versus model complexity (and computational cost).**

(3) Qualitative difference between $Plefka2[t]$ and $Plefka[t]$ & $[t-1]$ in practical applications

The authors quantitatively showed in Fig. 4 that $Plefka2$ is the most accurate model for estimating H and J . I can of course understand $Plefka2$ provides the least errors compared with other methods, but I cannot intuitively tell how significant this reduction of errors is in practical applications. For example, as written in Discussion, Ising models have been used to tell whether a system operates near a critical regime. In this application, how much difference would $Plefka2$ make in estimating critical points compared with the case where other $Plefka$ expansions are used?

We agree with the reviewer that we must present the merit of the different approximations in practical applications other than the simple inverse problem. In particular, the reviewer mentioned the criticality analysis as a practical application. Following this suggestion, **we performed the analysis of criticality from the data by combining the forward and inverse Ising problem as follows, and added this result in the Results section.** Thus we show how our methods can be used to examine if the nonequilibrium system is near a maximum fluctuation point or not.

We estimated the parameters H , J of the models from by the inverse Ising problem the data sampled at the critical temperature ($\beta = \beta_c$). Next, we used those parameters for computing the forward approximation, introducing a virtual inverse temperature $\tilde{\beta}$. Using this method, we computed both the average covariances in the system ($\langle C_{t_ik} \rangle$ and $\langle D_{t_il} \rangle$) and the entropy production of the system, to characterize the transition with rescaling of $\tilde{\beta}$. We then examined if the observed parameters yield the maximum fluctuations and entropy production compared to other conditions realized by the rescaled parameters. The result is included at the end of the results section and in a new figure (Fig. 5), in which the new method $Plefka2[t]$ overall performs better than the others.

This addresses the question on “how much difference would $Plefka2$ make in estimating critical points compared with the case where other $Plefka$ expansions are used”. We added this analysis in the Result section of the revised manuscript.

Furthermore, the authors stated that $Plefka[t]$ and $Plefka[t-1]$ actually “does not converge”. I do not exactly understand what this “does not converge” mean but I interpret this as the fact that we cannot basically use $Plefka[t]$ and $Plefka[t-1]$ as a valid estimation method. Is that correct? If so, I think this is a huge qualitative difference between $Plefka2[t]$ and $Plefka[t]$ & $[t-1]$ and the authors should emphasize this fact more to strengthen the advantage of $Plefka2[t]$. Interestingly, $Plefka[t,t-1]$ is better than $Plefka[t]$ & $[t-1]$ although $Plefka[t,t-1]$ is the simplest and the most crude approximation, which also probably means that there are some practical flaws in $Plefka[t]$ & $[t-1]$. I would like the authors to clarify and discuss this point more.

Motivated by the inquiry of the reviewer, we inspected in detail the convergence problems of $Plefka[t]$ and $Plefka[t-1]$. The methods did not converge in the sense that use of Boltzmann learning equations (materials and methods) with approximated correlations and means did not minimize the error between the data and the models. We found that this problem was a combination of 1) using covariances instead of correlations in the Boltzmann learning equations (Eq. 53 of the previous manuscript) and 2) using a small sample size (this was not a problem in the forward approximation, but it is in the inverse since the algorithm interacts recursively with the data). After fixing these numerical errors now $Plefka[t]$ and $Plefka[t-1]$ converge correctly, they show a precision that is similar to $Plefka2[t]$. **We revised the section of inverse Ising problem with these new results.** Thus in the revised manuscript, we no longer claim the superiority of $Plefka2[t]$ for the inverse Ising problem. However, we note that $Plefka2[t]$ performs better in the newly introduced analysis on the phase transition, and that the major goal of this manuscript is to provide an unified view on the different mean-field theories by which we can systematically propose new approximation methods.

Reviewer #2 (Remarks to the Author):

The paper introduces an unifying framework for mean field approximations to the dynamics of asymmetric kinetic Ising systems. While the paper is interesting and the contribution has potential, the writing has important flaws that make it hard to judge the content properly. Below, I provide a list of my main concerns, followed by a number of suggestions.

Main concerns

1. The paper seems to be written for a highly specialised audience, and hence unless a substantial changes are done it might not fit a generalist journal like Nature Communications. In particular, there is not enough argumentation of why non-specialists should care about the kinetic Ising model, mean field approximations, or the contribution proposed by the authors. Furthermore, the paper focuses on technicalities of the method but do not provide applications to real data that could provide evidence of its value for people who could want to use the model. As it stands, the paper fails to effectively speak to a larger audience — beyond people already invested in kinetic Ising models.

We agree with the reviewer that the manuscript has to be framed into broader contexts. We truly appreciate this suggestion because we believe that we made the introduction much more appealing to the general audience by addressing this concern.

In short,

- 1. We added importance of having unified analytical methods to examine non-equilibrium (irreversible) systems in the Introduction.**

2. **We emphasized applications of the approximation methods for investigating networks near-maximum fluctuation regime. In particular, we added a new practical method and result for examining if the system is at the regime of the maximum fluctuation and irreversibility.**
3. **We added discussion on the contribution of the presented mean-field methods to other fields (machine learning and social systems) in the Introduction.**

Below we explain these changes.

1. Importance of having unified analytical methods to examine non-equilibrium systems in the Introduction

First, we strengthened the arguments on the pressing need for analyzing large, nonequilibrium systems, and the importance of having unified practical analysis frameworks for the nonequilibrium systems near criticality in Introduction.

We apologize that we did not properly introduce the definition of nonequilibrium processes, and we only briefly introduced recent progress on the second law of thermodynamics at the end of the Discussion in the previous manuscript. Therefore, this important message might not be appreciated by the Reviewer, which makes s/he suggest to remove arguments on the nonequilibrium systems and replace it with non-stationarity (Suggestion #1).

A system is away from equilibrium if it is driven by past activities of the system's elements and it undergoes irreversibility with respect to time: Namely, if the probabilities of a forward path and backward path are, on average, different (being this time-asymmetry governed by the fluctuation theorem, from which the second law of thermodynamics can be derived [1-7]). However, despite the recent theoretical developments on the nonequilibrium processes in the information thermodynamics, we still need the practical analysis methods that make it possible to analyze particular nonequilibrium systems.

The kinetic Ising system is, to our knowledge, one of the most studied prototypical nonequilibrium system. It is driven by past activities. With symmetric couplings it recovers the classical equilibrium Ising model. Otherwise, the process is irreversible and display nonequilibrium dynamics. Note that we use non-symmetric couplings in the model (meaning that the connection from $s_{\{j,t-1\}}$ to $s_{\{i,t\}}$ is different than $s_{\{i,t-1\}}$ to $s_{\{j,t\}}$). As this asymmetric Ising model is hard to solve analytically, the presented framework provides an interesting opportunity for the study of large nonequilibrium networks.

In the revised manuscript, we introduced the definition of a nonequilibrium system with selected references to the recent development of information thermodynamics, and described how the proposed model displays nonequilibrium dynamics.

[1] D. J. Evans, E. G. D. Cohen, and G. P. Morriss, Phys. Rev. Lett. 71, 2401 (1993).

[2] C Jarzynski. Phys Rev Lett 78, 2690 (1997)

- [3] G. E. Crooks, J. Stat. Phys. 90, 1481 (1998).
- [4] G E Crooks. Phys Rev E 60, 2721–26. (1999)
- [5] J. L. Lebowitz and H. Spohn, J. Stat. Phys. 95, 333 (1999).
- [6] D. J. Evans and D. J. Searles, Adv. Phys. 51, 1529 (2002).
- [7] U Seifert Reports on Progress in Physics 75, 126001 (2012)

2. Emphasis on applications of the approximation methods for reconstruction of networks near-maximum fluctuation regime

In addition, we aimed at presenting the approximation methods as practical tools to analyze biological or social phenomena. Many biological and social systems are governed by the non-equilibrium dynamics rather than equilibrium ones. Furthermore, they often operate near a maximum fluctuation regime (an analogue of critical conditions for equilibrium systems). We performed the analysis at the challenging aspects of the nonequilibrium system expected in biological and social systems.

First, in the forward and inverse Ising problem, we chose the parameters that make the kinetic Ising system display maximum fluctuations. This point was found by the replica method (SI 6), and confirmed by numerical simulation (SI. Fig. S2). We demonstrated that the new Plefka2[t] performs better than any other methods at this regime for forward Ising problems (Fig.3, 4). **In Introduction of the revised manuscript, we stressed that we performed the analysis at the nonequilibrium system at near-maximum fluctuation regimes.** Second, to further strengthen the usefulness of the methods, **we added an analysis of the transition around the maximum fluctuations point in the Results section by which users can examine if the observed dynamics is near a critical regime or not.**

In addition to the above, we focused on predicting the transient dynamics of the correlations in the forward Ising problems. The nonequilibrium processes may undergo the steady-state (nonequilibrium steady-state, NESS), and can show transient dynamics toward the steady states. In the manuscript, to demonstrate the practical merit of these methods, we decided to predict the future correlations when the system undergoes the transient dynamics because it is even more challenging than estimating correlations at the steady states. We demonstrated that our Plefka[2] performs better in these tasks (Fig.3B,C; Fig.4). However, please note that this is an additional merit of the proposed method.

3. Contribution of the mean-field approaches to machine learning communities in the Introduction

Lastly, the kinetic Ising model and its mean-field approximation are potentially used in machine learning applications, because the kinetic Ising model with symmetric connections realizes the dynamics of the Boltzmann machine heavily used in the machine learning community. This manuscript provided the novel approach (Plefka2[t]) on the inverse problem; learning association of binary patterns from data. Since Plefka2[t] provides better estimation, it serves as

a better learning algorithm than the other. **We added a sentence referring to machine learning in the Introduction. In addition, we added references to applications of the kinetic Ising models in the financial market.**

Finally, although we agree that adding analysis of the real data would provide value to the manuscript, we are afraid that it would be an iteration of the simulation analysis, without verification of the method due to the lack of knowledge on the ground truth. **In order to add real values for the people who would like to use the method, we added a guideline based on the trade-off between accuracy of the method and computation time so that people can choose the best method in their problem (SI.7).** We will also provide the code and a link to the website introducing how to use the code, which was added as "Data Availability".

We appreciate the reviewer that makes us reconsider these important points to make this paper appealing to a wider audience. We included these arguments in the revised manuscript as stated at the beginning of this reply.

2. It is not clear which is exactly the contribution of the paper. It seems that some of these mean-field approximations existed beforehand and some others didn't, but this is not explicitly clarified. Also, how relevant is the asymmetry for the contribution? Please clarify all these issues, explaining why this is relevant for a generalist audience.

In order to clarify the contribution of the paper, **we added a paragraph describing the structure of the paper as the last paragraph of Introduction.** In addition, we made it clear that the major goal of the paper is to present a unifying framework in which we can systematically propose new approximations that are more accurate. We also made it clear that P₂ is our proposed novel method, **we also rewrote the introduction (the first and second paragraphs) of the P₂ method, as**

"The proposed framework is also a powerful tool to develop novel P₂ expansions. Here, we propose new extended models that directly approximate pairwise activities of the units."

We hope that the question regarding the importance of considering asymmetry (in couplings) was clarified by our reply to the Major concern 1.

3. While the authors claim that their method is based on information geometry, I failed to see how exactly information geometry is driving their approach. From what I see in pg.3, P_α is just another 1-dim manifold that passes through P and Q, but is not clear if it is also a geodesic is some metric. Moreover, I don't see specific tools of information geometry at play in the derivation or application of the method. Given the importance that the authors place on

information geometry when describing their framework, the authors should develop more this aspect of the work.

We agree with the reviewer that P_α is not a geodesic based on the KL metric. However, the framework is based on finding the projected model Q from P under the minimization principle of $D[P||Q]$ (Eq.8). This is a point found by m -projection which is mathematically intractable. **In the revised manuscript, we rewrote the section of "geometrical approaches to the Plefka expansion", and more clearly motivated the introduction of P_α as a tractable alternative to this m -projection. In addition, we added a paragraph that discusses the relation of our approach to the variational approach that is based on $D[Q||P]$, and clarified which divergence ($D[P||Q]$ or $D[Q||P]$) was used to derive the mean-field equations in other works that utilizes the information geometry (Bhattacharyya & Keerthi, 2000, Amari et al, 1992, Tanaka, 1998).** These comparisons clearly rely on the formulation of our work into the framework of information geometry.

Suggestions

1. In the beginning of the introduction, the authors argue about the importance of non-equilibrium dynamics. However, it is not clear if they are really talking about non-equilibrium, in a thermodynamic system, or just talking about non-stationary. As I don't see any real thermodynamics at play through the paper (i.e. work, heat, etc), I suggest to modify the discussion and state it in terms of non-stationarity, this to avoid confusion and misinterpretations.

We believe that the previous version of the paper did not explain with enough details to bring the justification and importance of non-equilibrium properties in our model. The mean-field equations derived in the paper are generalized for the Ising models with asymmetric couplings. This is a sufficient condition for breaking detailed balance and having entropy production in the dynamical Ising network.

Once again, the system is away from equilibrium conditions if it undergoes irreversibility with respect to time. If the probabilities of a forward path and backward path are, on average, different. According to the fluctuation theorem (FT) developed by Evans, Jarzynski, Crooks and others, the second law of thermodynamics is derived as the entropy production caused by this time-asymmetry, which is assessed as the KL divergence between the forward and backward paths [1-5]. Along this line, according to [7] "Going beyond the thermodynamic framework, it turns out that many of the FTs hold formally true for any kind of Markovian stochastic dynamics. The thermodynamic interpretation of the involved quantities as heat and work is not mandatory to derive such a priori surprising relationships between functionals defined along dynamic trajectories." Furthermore, although the current manuscript does not consider the exogenous effect, the framework has been extended to include the effect of exogenous inputs on the

process. Although we do not deal with work or heat explicitly, the work done to the system can be expressed by changes in the parameters of the Ising system, and heat is represented as the entropy production by the irreversible process.

In the new version of the text, we have justified the importance of non-equilibrium and the generalization to asymmetric Ising model in the introduction and included relevant references.

2. Please add some references when first introducing the kinetic Ising model in the introduction.

We introduced references to Hertz et al, 2013, Roudi et al, 2015 when we first introduced the kinetic Ising model.

- Hertz, J., Roudi, Y., & Tyrcha, J. (2013). Ising model for inferring network structure from spike data: In *Principals of Neural Coding*.
- Roudi, Y., Dunn, B., & Hertz, J. (2015). Multi-neuronal activity and functional connectivity in cell assemblies. *Current opinion in neurobiology*, 32, 38-44.

3. It would be appreciated to have a bit more of discussion about the limitations of direct fitting kinetic Ising models in real data. Is it just computationally expensive, or plainly intractable? Which kind of sizes are still workable? (considering that nowadays some companies are working with statistical models of hundreds of billions or parameters). This discussion is important, as it provides the reason of existence of the approximations that motivate this work.

Fitting the equilibrium Ising model is computationally intractable because one needs to use probabilities of the 2^N patterns to compute expectation parameters at each iteration. In contrast, fitting the kinetic Ising model to real data is not computationally intractable but the complexity only linearly increases with N because of the conditional independence among the neurons. However, one needs to compute the average over all trials and time steps ($R \times T$) at each iteration, which is computationally expensive. By the mean-field approximation, we could remove the average computation ($R \times T$ times the number of iterations), which significantly reduce the computational time. **In the revised manuscript, we emphasized the reduction of computational time in the Materials and Methods section.**

As for the the number of workable size, **in the Introduction of the revised manuscript, we referred to some examples in the literature showing that use of the models to infer data (without approximations) fits to networks of around a hundred of neurons (e.g. Tyrcha et al 2013, Tkačik et al, 2015).**

4. When discussing symmetry and asymmetry in kinetic Ising models, it is not clear what kind of symmetry the authors are referring to. Please clarify this, and explain why this is important for theoretical and also practical applications.

Asymmetry refers to the coupling in the system. **We clarified it in the revised manuscript. When we introduced the kinetic Ising model, we wrote**

"When the couplings are asymmetric (i.e., $J_{ij} \neq J_{ji}$), the system is away from equilibrium because the process is irreversible with respect to time."

We hope that its importance is clarified in the revised manuscript as well as by the reply to Major concern 1 about the non-equilibrium system and detailed balance.

5. In the section "The Kinetic Ising Model", the model is not clearly specified. For example, Eq. (1) specify the conditional probability of individual units, and then Eq. (3) uses the joint conditional probability of the whole system. How are these two related? Is the joint conditional equal to the product of the individual conditionals? Also, it is assumed that the temporal dynamics are Markovian (i.e. that the system's state at time t do not depend to past states $t-2$, $t-3$... given the state at $t-1$). Finally, shouldn't an initial condition be specified in order to have a well-defined stochastic process?

Thank you very much for these comments we modified the description of the model section, including:

1. **We modified equation (1) so that now it describes the joint conditional probability of the system (which is the product of individual conditional probabilities).**
2. **We mentioned explicitly that the system exhibits Markovian dynamics in Introduction.**
3. **We clarified the trajectories depend on an initial probability distribution as follows.**
 - a. ", and trace the evolution of the system from the initial conditions $P(s_0)$." after Eq.2.
 - b. " while the marginalization was taken over the initial states of the path using $P(s_{t-k})$." in the paragraph before the subsection on Plekfa[t,t].

6. I don't see a need of mention Plefka expansions in pg.2.

Unfortunately, we could not understand this suggestion. The Plefka expansion is a central topic in our paper, and needs to be introduced in Introduction. Therefore, we keep the description. It first appears in pg.2. However, as it is the first time Plefka expansions are mentioned we tried to provide a better description.

7. When introducing the KL divergence in Eq. (8), it is not clear it is the relevant metric to be used. There are many non-negative divergences that quantify “distances” between two joint probabilities, so, why the KL is the relevant one here? Please provide a motivation there.

We agree that the use of the KL divergence was not well motivated. In the context of Plefka expansions, it has been shown that minimizing the KL divergence is equivalent to minimizing the Gibbs free energy of the reference model. Minimization of $D(P||Q)$ with respect to the independent manifold Q leads to the point on this manifold with the first moment equal to the original distribution. Similarly for Plefka2[t], we find that in addition to the first moment the second moments of the projection point are equal to the second moments of the original model. Since the goal is to approximate these moments, we believe that the KL divergence $D(P||Q)$ is the most appropriate metric for this analysis. This can be better understood if we think for example about using $D(Q||P)$ instead of $D(P||Q)$. Minimizing $D(Q||P)$ leads to the e-projection from P to the reference manifold (which corresponds to the naive mean-field approximation) and introduces a bias in approximation of the moments m_t of $P(s_t)$ [10]. The same is true for other alpha-divergences [11]. **In the new version of the manuscript, we revised the subsection of “Geometrical approaches to the Plefka expansion”, and motivated the use of $D(P||Q)$ as follows:**

“Our goal is to find the average activation rates of the target distribution $P(\{s_t\} | \{H, \{J\})$. As shown by information geometry \cite{tanaka_information_2001}, this can be achieved as the orthogonal projection of the target distribution to the sub-manifold \mathcal{Q}_t .”

This orthogonal projection is obtained by by minimizing the Kullback-Leibler (KL) divergence from $P(\{s_t\})$ to $Q(\{s_t\})$.”

[10] T. Tanaka, Information Geometry of Mean-Field Approximation, in Advanced Mean Field Methods: Theory and Practice(MIT Press, 2001).

[11] S.-i. Amari, S. Ikeda, and H. Shimokawa, Information geometry of α -projection in mean-field approximation,inAdvanced Mean Field Methods: Theory and Practice(MIT Press, 2001).

8. The Plefka approximation is mentioned many times before it is defined. Also, it feels awkward that something so important for the paper is defined within an example.

In the revised manuscript, we moved the Plefka approximation into the general introduction of our approach. Therefore, now it is written before we explain the example of Plefka[t-1,t]. **Also, we revised the brief definition of the Plefka approximation when it first appeared in Introduction.**

9. In the example provided in pg.4, it is not clear what the reader should focus in, or why it should care. Please provide more description of what is the purpose of the example.

We agree with the reviewer that we did not motivate this example of Plefka[t-1,t] properly in the previous manuscript. This example derives the classical 1st and 2nd order mean field equations, thus it illustrates how widespread approximation methods are obtained within our framework. **We have introduced a paragraph motivating and justifying the example at the beginning of this section as:**

"Before elaborating different mean field approximations, it is instructive to further explain and clarify our method by deriving the known results of the classical nMF and TAP approximations for the kinetic Ising model \cite{kappen_mean_2000, roudi_dynamical_2011}."

10. The caption of Fig.2 fails to provide a clear explanation of the point of the figure, while calling a number of references. This might appeal an specialist but not to the general audience.

We rewrote the caption trying to make it intuitive for a general audience. However we would like to ask in favour of keeping the references because we think that it is important to give easy access to the original papers that the graphs are referring to.

11. Figure 2 seems to have something odd with the labels, as the last figure has no letter (should it be F?), and the name in front of E is way away of it.

The two last figures correspond to label E. **We redesigned the label position to make this clearer.**

Reviewer #3 (Remarks to the Author):

The paper unifies and extends various existing mean-field approximations of the kinetic Ising model and presents in addition new approximations based on the Plefka expansion. The various approaches are carefully compared and discussed.

This work is to my knowledge novel, and I think that the manuscript is valid and worth of publication in NATURE Communications.

However, one issue could be clarified (or at least mentioned as an unsolved problem) by the authors:

The validity of the static TAP equations is restricted in the spin glass regime according to [22]. A specific, but important example for an incorrect TAP solution in the zero field situation is the paramagnetic solution with vanishing local magnetisations below the spin glass temperature.

Similar restrictions are expected for the dynamical case. The question is what is going on, if the dynamics enters the prohibited region of the statics. To the best of my knowledge, these problems have not been worked out for mean field approximations of the kinetic Ising systems

We appreciate the reviewer for giving us expert knowledge. We agree with the reviewer's point of view. In our paper, we do not inspect the spin-glass transition when we compare performance of the approximation methods. Instead we explore an order-disorder transition, around which is still challenging for the existing approximations. We chose this point because we could analytically obtain the transition point by the replica method.

Exploration of the spin-glass transitions is of course important. However,, it is difficult to inspect analytically since replica symmetric methods do not work and solutions need complex replica-breaking assumptions. Therefore, we should like to make it a future research topic. **we introduced this issue in the discussion as future work that could follow this publication as:**

" note that low-temperature spin phases (e.g. spin-glass phases) also impose limitations on mean-field approximations \cite{plefka_convergence_1982, mezard_exact_2011}, which could be further explored with methods like the ones presented here."

As a minor remark I recommend to enlarge Fig. 3 and Fig.4. In the present form they are tiny.

With these amendments I thus recommend this work for publication.

Thank you very much. We agree that the figures are small and it is not easy to introduce them in a compact form. **We enlarged the figures and will make sure that the details of Figures 3 and 4 are visible at the production stage.**

REVIEWERS' COMMENTS

Reviewer #2 (Remarks to the Author):

The authors have done a great work in responding my questions and improving the manuscript. I'm happy to recommend the manuscript for publications.

That being said, there are still some minor points that would be helpful to address before publication:

- In general, is not trivial for a non-expert reader to see where the novel content starts. Could it be possible to separate old and new in different sections?
- Please revise the second paragraph of the introduction, which doesn't read as smoothly as the rest of the introduction.
- The intro says that non-equilibrium behaviour happens when parameters such as external fields change. What happens when these changes are so slow that the system can equilibrate? These quasi static systems remain in equilibrium, unless I am missing something?
- I am not entirely sure if the models with dependency of past activity such as the ones in Ref.[19] can be said to be out-of-equilibrium models, as they seem to satisfy detailed balance. Moreover, I think readers may appreciate some discussion between the ideas of models that account for temporal relationships, models that are non-stationary, and models that correspond to non-equilibrium processes. Personally, I found problematic to talk about non-equilibrium when in the absence of a clear notion of "energy" (such in neural data), but I also can see the appeal of other opinions on this. Despite the position taken on this, I think a discussion would be helpful.
- When introducing the model in Eqs. (1) and (2), it would be relevant to clarify that Markov dynamics are assumed by definition.
- When introducing the features of interest (eqs. 4-6), it is not clear why those are the relevant features and not others. Please comment a bit on this.
- Section "Geometric approaches..." could use of more subsections: one for generalities (starting at line 202), and one for the pleka expansion (starting at line 237).
- In eq. 9 are related, I'd suggest to use Q^* (e.g. as argument of the KL) to clarify that this is not valid to any Q but only for the optimal one. Also, after Eq.9 it would be nice to discuss a bit what may happen with the other observables (eg (5) and (6)).

Finally, as a side note, maybe this recent preprint (<https://arxiv.org/abs/2005.02526>) could be of interest. It shows interesting results about entropy production in fMRI data, but computes entropy production in a questionable manner. I wonder in the methods presented here could serve to carry out similar analyses in a more rigorous fashion.

Reviewer #3 (Remarks to the Author):

The new version fulfills my requirements.
It can be published in the present form.

Response to reviewers

Reviewer #2 (Remarks to the Author):

The authors have done a great work in responding my questions and improving the manuscript. I'm happy to recommend the manuscript for publications.

That being said, there are still some minor points that would be helpful to address before publication:

- In general, is not trivial for a non-expert reader to see where the novel content starts. Could it be possible to separate old and new in different sections?

Following the editor indications, we have englobated all contributions under a 'Results' section. We think this may clarify where the review of previous work ends.

- Please revise the second paragraph of the introduction, which doesn't read as smoothly as the rest of the introduction.

We have reviewed this paragraph.

- The intro says that non-equilibrium behaviour happens when parameters such as external fields change. What happens when these changes are so slow that the system can equilibrate? These quasi static systems remain in equilibrium, unless I am missing something?

We appreciate the comment. We have indicated that it is just the case of rapidly changing external fields what cause non-equilibrium conditions.

- I am not entirely sure if the models with dependency of past activity such as the ones in Ref.[19] can be said to be out-of-equilibrium models, as they seem to satisfy detailed balance. Moreover, I think readers may appreciate some discussion between the ideas of models that account for temporal relationships, models that are non-stationary, and models that correspond to non-equilibrium processes. Personally, I found problematic to talk about non-equilibrium when in the absence of a clear notion of “energy” (such in neural data), but I also can see the appeal of other opinions on this. Despite the position taken on this, I think a discussion would be helpful.

We thank the reviewer for spotting this. We did not realise that the kinetic models in Ref.19 assumed detailed balance. We have removed that reference and substituted by other two that present models that do not satisfy detailed balance (Tyrcha et al 2013, Hertz et al, 2013).

We understand the worry about the absence of an explicit description of energy for modelling non-equilibrium processes. Of course we consider that non-equilibrium phenomena arise from the dissipation of energy, which results in a time-asymmetry in their evolution. Stochastic thermodynamics provides a formulation of this asymmetry in terms of entropy production. In our paper, we address systems (representing spike trains in neural networks) displaying this asymmetry, which are modelled by introducing asymmetry in the causal mechanisms (Ising couplings) of a Markov chain. As well, we quantify time-asymmetry by measuring entropy production in the system. Here, our view is that we do not model energy dissipation explicitly, but that the time-asymmetry found in our models can only be displayed by physical systems in which energy dissipation takes place. We have tried to reference this when we introduce the idea of non-equilibrium in our model.

- When introducing the model in Eqs. (1) and (2), it would be relevant to clarify that Markov dynamics are assumed by definition.

We have specified that the system behaves as a discrete-time Markov chain.

- When introducing the features of interest (eqs. 4-6), it is not clear why those are the relevant features and not others. Please comment a bit on this.

We added the following justification “Note that $\langle m_t \rangle$ and $\langle D_t \rangle$ are sufficient statistics of the kinetic Ising model used here (because it is defined as a maximum caliber model). Also they used in solving the inverse Ising problem (see Materials and Methods). We find it suitable to add same-time correlations $\langle C_t \rangle$ as they are simple to measure and commonly used to describe neural activity and are used by some of the mean field approximations in the literature (e.g. Roudi and Hertz, 2011).”

- Section “Geometric approaches...” could use of more subsections: one for generalities (starting at line 202), and one for the Plefka expansion (starting at line 237).

We have divided the subsection in two (the journal does not allow subsubsections): “Geometrical approach to mean-field approximation” and “The Plefka expansion”.

- In eq. 9 are related, I’d suggest to use Q^* (e.g. as argument of the KL) to clarify that this is not valid to any Q but only for the optimal one. Also, after Eq.9 it would be nice to discuss a bit what may happen with the other observables (eg (5) and (6)).

We changed the equation to make it clear that the derivative is evaluated at Q^*

We briefly mention that in this case the approximation is independent of the other observables (5) and (6), and that later in the paper we will consider approximations that depend on their values.

Finally, as a side note, maybe this recent preprint (<https://arxiv.org/abs/2005.02526>) could be of interest. It shows interesting results about entropy production in fMRI data, but computes entropy production in a questionable manner. I wonder in the methods presented here could serve to carry out similar analyses in a more rigorous fashion.

We appreciate the suggestion. We have included a comment and the reference at the end of the discussion section.

Reviewer #3 (Remarks to the Author):

The new version fulfills my requirements.
It can be published in the present form.